# Prime editing efficiency and fidelity are enhanced in the absence of mismatch repair

J. Ferreira da Silva[1,2,4], G. P. Oliveira[1,4], E. A. Arasa-Verge [1], C. Kagiou[1,2], A. Moretton[1,2], G. Timelthaler[1], J. Jiricny[3] & J. I. Loizou [1,2✉]

Prime editing (PE) is a powerful genome engineering approach that enables the introduction of base substitutions, insertions and deletions into any given genomic locus. However, the efficiency of PE varies widely and depends not only on the genomic region targeted, but also on the genetic background of the edited cell. Here, to determine which cellular factors affect PE efficiency, we carry out a focused genetic screen targeting 32 DNA repair factors, spanning all reported repair pathways. We show that, depending on cell line and type of edit, ablation of mismatch repair (MMR) affords a 2–17 fold increase in PE efficiency, across several human cell lines, types of edits and genomic loci. The accumulation of the key MMR factors MLH1 and MSH2 at PE sites argues for direct involvement of MMR in PE control. Our results shed new light on the mechanism of PE and suggest how its efficiency might be optimised.

[1] Institute of Cancer Research, Department of Medicine I, Comprehensive Cancer Centre, Medical University of Vienna, Borschkegasse 8a, 1090 Vienna, Austria. [2] CeMM Research Center for Molecular Medicine of the Austrian Academy of Sciences, Vienna, Austria. [3] Institute of Biochemistry of the ETH Zurich, Otto-Stern-Weg 3, 8093 Zurich, Switzerland. [4] These authors contributed equally: J. Ferreira da Silva, G. P. Oliveira.
✉email: joanna.loizou@meduniwien.ac.at

CRISPR-Cas9-based genome editing technologies are powerful new tools of functional genomics, with considerable potential as future therapeutics[1]. However, the efficiency of currently available genome editing protocols is limited. Moreover, the process gives rise to undesirable side products that hinder the implementation of this technology in clinical settings. To overcome these hurdles, there is need to identify the DNA metabolic pathways and molecular mechanisms that govern editing outcomes, as well as the activities of these pathways in different cellular and tissue contexts[2–5]. The first generation of Cas9-based genome engineering tools used nucleases that could be directed to any desired region of the genome by a single-guide RNA (sgRNA). Following the targeting of a site-specific DNA double-strand break (DSB), the endogenous DNA end-joining pathways frequently repair this lesion in an error prone manner, leading to insertions or deletions (indels) that give rise to loss-of-function alleles[6]. This approach was further adapted to include either a single-stranded or double-stranded donor template containing the desired edit. Here, the DSB is processed by homology-directed repair (HDR), which catalyses the insertion of the donor template that includes the edit. Unlike the former approach, which generates random indels, the latter method permits the introduction of desired indels, as well as point mutations, into the genome[3]. However, since HDR is inefficient, depends on potentially deleterious DSBs and requires cell division, alternative approaches were needed.

Amongst such alternative approaches are base editing (BE) and prime editing (PE). The former uses nucleobase modification chemistry to efficiently and precisely incorporate single nucleotide variants into the genome of cells[7–9], but its scope is limited to single-base substitutions. This led to the development of PE as a highly versatile genome editing approach that allows for the targeted insertion of indels, point mutations and combinations thereof into the genome[10]. PE utilises a fusion of a Cas9(H840A) nickase[1] and reverse transcriptase (RT) that is targeted to a precise genomic region by a PE guide RNA (pegRNA). The pegRNA includes the desired sequence change, as well as a short 3′ terminal extension complementary to the 5′ sequence upstream from the nick within the target site. Annealing of the 3′ terminus of the pegRNA to the 3′ segment of the nicked DNA strand generates a substrate for the RT, which copies the RNA template and thus incorporates the desired edit into the 3′ extension of the nick. Dissociation of the RNA and annealing of the DNA strands generates a 3′ flap containing the edit. Transient melting and reannealing of the nicked target site give rise to a mixture of molecules containing either 3′ or 5′ flaps. Successful installation of the desired edit requires removal of the 5′ flap and ligation of the resulting nick to yield a DNA heteroduplex containing the edit in the RT-synthesised strand. The editing outcome of this method, referred to as PE2, depends on the resolution of this heteroduplex. Utilising an additional sgRNA that directs nicking to the original DNA strand, either concurrently to the edit installation (PE3), or subsequently (PE3b), increases PE efficiency[10]. The increased efficiency in PE3 strategies has been suggested to require the DNA repair pathway known as DNA mismatch repair (MMR) that would function in the repair of the nicked, non-edited strand, by utilising the edited strand as template[11,12].

Due to its versatility, PE has been used in a wide variety of models, such as zebrafish[11], rice and wheat[13], mouse[14], and human stem cells[15]. A notable feature of PE is its highly variable rates across different genetic backgrounds, even within the same genomic locus and using the same pegRNA[10]. To address whether this could be explained by different DNA repair capacities, we performed a targeted genetic screen aimed at identifying DNA repair factors involved in PE. Here, we uncover an inhibitory role

for the MMR pathway in PE and show that MMR proteins localise to sites of PE to directly counteract edit installation, rather than promote it. Thus, deletion or transient depletion of MMR factors increase PE efficiency and fidelity across different edit sites, types and cell lines.

## Results

**A targeted genetic screen identifies the DNA repair pathway mismatch repair as inhibitory for prime editing.** To investigate the DNA repair requirements for PE, we conducted a targeted genetic screen, utilising a collection of isogenic knockouts in the human near-haploid HAP1 cell line (Supplementary Data 1). The 32 targeted genes were selected to represent divergent functions within all known human DNA repair pathways. The library thus provided a comprehensive coverage of the DNA damage response. The cell lines received the PE machinery, including the Cas9(H840A)-RT and a pegRNA encoding a 5-base pair (bp) deletion in the HEK3 locus. PE efficiency was determined by amplicon sequencing of the genomic locus.

Wild-type HAP1 cells were remarkably inefficient at PE (<1% alleles edited). In contrast, isogenic HAP1 cell lines mutated at the MLH1, PMS2, MSH2, EXO1, and MSH3 loci displayed higher PE levels, ranging from 2-fold to 6.8-fold (Fig. 1a). Disruption of other DNA repair pathways had little or no impact on PE efficiency. This finding clearly indicates that MMR functions to inhibit PE. Of all MMR genes targeted in the screen, only the loss of MSH6 failed to increase PE efficiency.

The MMR pathway evolved to correct base/base mispairs and small indels arising in DNA during replication and recombination. To initiate repair, these lesions are recognised by the heterodimers MutSα (MSH2–MSH6) or MutSβ (MSH2–MSH3). Whereas MutSα recognises base/base mismatches and indels of 1–2 nucleotides, larger indels are recognised by MutSβ[16–20]. Substrate binding brings about an ATP-dependent conformational change of the MutS complexes and recruitment of the MutLα (MLH1-PMS2)[21] or MutLβ (MLH1–MLH3)[22] heterodimers. Assembly of the MutL complex together with RFC and PCNA[23], bound at a pre-existing strand discontinuity (either a nick or a free 3′ terminus), activates cryptic endonuclease activity of the PMS2 or MLH3 proteins, which then introduce additional DNA single-strand breaks (SSBs) into the discontinuous DNA strand, in the vicinity of the mismatch. These SSBs act as entry points for EXO1, which degrades the discontinuous strand in a 5′ to 3′ direction up to, and some distance past, the misincorporated nucleotide(s)[24]. The resulting gap is filled-in by DNA polymerase δ and the remaining nick is ligated by DNA ligase I[25–27]. Since the edit introduced in our screen is a 5 bp deletion, this makes it a substrate of MutSβ, but not MutSα[18], which explains the lack of an effect on editing upon the loss of MSH6 (Fig. 1a). This result highlights the highly specialised nature of the DNA damage response that functions on different substrates.

**Mismatch repair hinders PE2 and PE3 across several human cell lines, genomic loci and edit types.** To further explore the inhibitory role of MMR in PE, we expanded our investigations to a panel of MMR-deficient human cell lines, alongside their complemented counterparts, in which we measured the editing efficiency and fidelity of the HEK3 locus. We used the colorectal cancer line HCT116, which is mutated in both MSH3 and MLH1, alongside the MMR-proficient HCT116 cell line complemented with chromosomes 5 and 3 that house the wild-type copies of the two genes, respectively (Supplementary Fig. 1A)[28,29]. The endometrial adenocarcinoma cell line HEC59, which is mutated at the MSH2 locus, was used together with its MMR-proficient counterpart complemented with chromosome 2 that carries the wild-

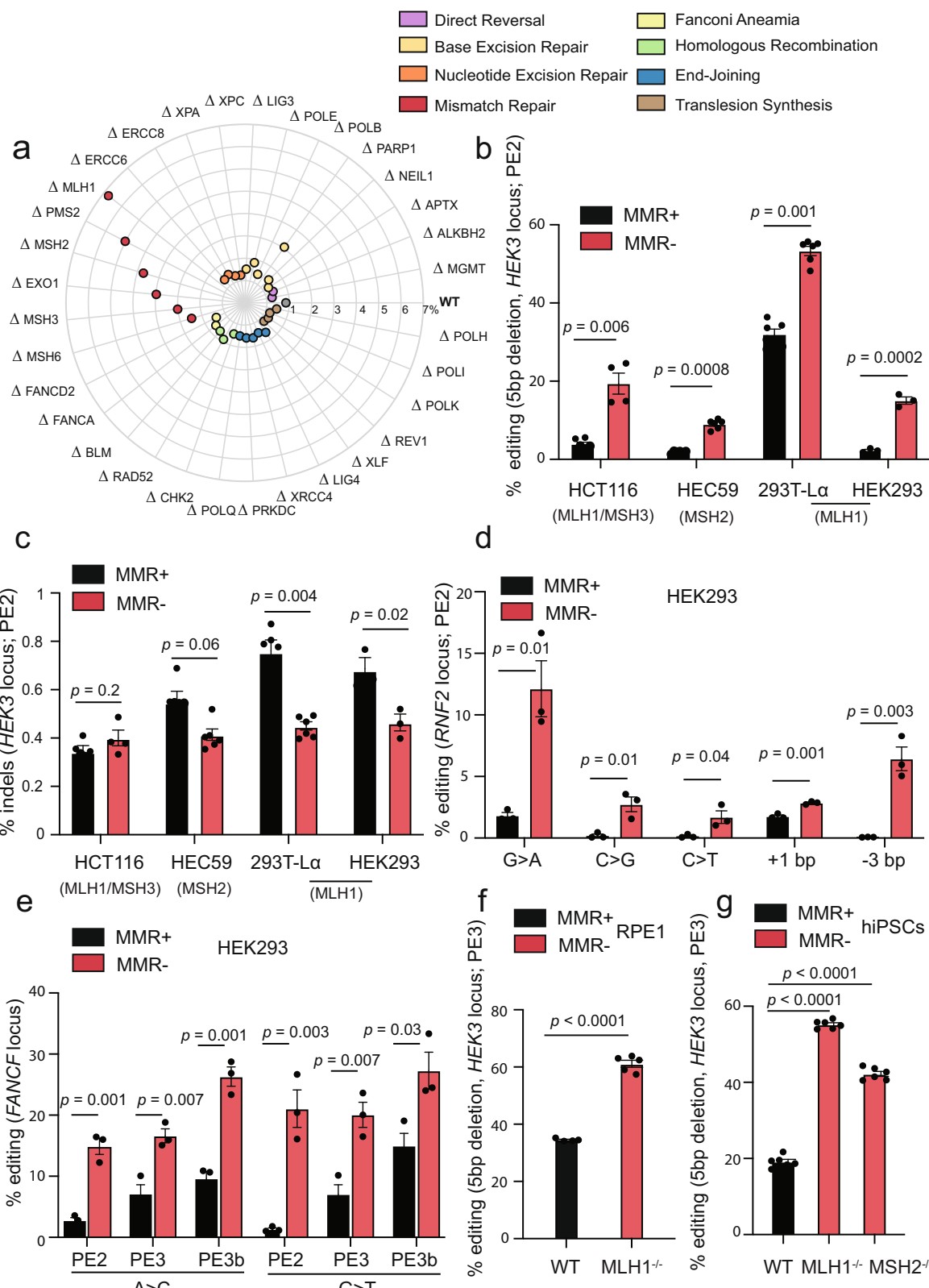

We controlled for the transfection efficiencies of all the matched MMR-deficient and proficient cell line pairs and showed that these were comparable, as measured by the percentage of cells transfected with a GFP expressing plasmid (Supplementary Fig. 1E). We then performed PE2 editing by deleting 5 bp within the HEK3 locus. All cell lines showed significantly increased PE2

type *MSH2* gene (Supplementary Fig. 1B)[30]. Additionally, we used a doxycycline-inducible model of MLH1 deficiency in the human embryonic kidney cell line HEK293T (293T-Lα) (Supplementary Fig. 1C)[31]. Finally, we generated an isogenic pair of MLH1 wild-type and knockout HEK293 cells (Supplementary Fig. 1D).

**Fig. 1 Mismatch repair inhibits prime editing in human cells. a** Genetic screen in 32 HAP1 isogenic knockout cell lines covering different DNA damage repair pathways, as well as their wild-type (WT) counterpart. Efficiency of installation of a five base pair (bp) deletion in the *HEK3* locus, using PE2 for $n = 2$ biologically independent experiments. Each radial line represents an increment of 1%. **b** PE2 of a 5 bp deletion in the *HEK3* locus in the indicated mismatch repair-deficient cell lines (MMR−), and their respective complemented counterparts (MMR+). For each cell line, the mutated MMR genes are represented. Editing efficiency measured for $n = 3$ biologically independent experiments (with technical replicates also depicted). **c** Percentage of indels after a 5 bp deletion in the *HEK3* locus using PE2 in varying mismatch repair-deficient (MMR−), and their respective complemented, cell lines (MMR+), for $n = 3$ biologically independent experiments (with technical replicates also depicted). For each cell line, the correspondent mutated MMR gene is indicated. **d** PE2 of the indicated types of edits (*RNF2* locus) in HEK293 cells wild-type (MMR+), or knockout for MLH1 (MMR−). Values correspond to editing efficiency, measured for $n = 3$ biologically independent experiments. **e** Efficiency of PE2, PE3, and PE3b after inducing A>C or G>T mutations in the *FANCF* locus. HEK293 wild-type (MMR+) and MLH1 knockout cells (MMR−) were used. Editing efficiency for $n = 3$ biologically independent experiments. **f** PE3 efficiency of a 5 bp deletion in the *HEK3* locus in RPE1 wild-type (WT) cells and an isogenic knockout MLH1 cell line (RPE1-MLH1$^{-/-}$), determined by Sanger sequencing and TIDE analysis, for $n = 3$ biologically independent experiments (technical replicates also depicted). RPE1 cells express Cas9(H840A)-RT in a constitutive manner (RPE1 PE2-BSD). **g** PE3 efficiency in wild-type (WT) human induced-pluripotent stem cells (hiPSCs), and isogenic knockouts for MLH1 and MSH2 (MLH1$^{-/-}$, MSH2$^{-/-}$), for $n = 3$ biologically independent experiments (technical replicates also depicted). Statistical analysis performed using unpaired two-tailed Student's *t*-test across biological replicates only. Error bars reflect mean ± s.e.m. Source data are provided as a Source Data file.

editing (ranging from 1.7-fold to 6.6-fold) when MMR was ablated, compared to their MMR-proficient counterparts (Fig. 1b). Importantly, even though PE efficiencies were increased by MMR deficiency, this did not come at the expense of higher indel frequencies within the amplicon region (Fig. 1c). Indeed, we observed that loss of MMR prevented unwanted indels at the *HEK3* locus in the 293T-Lα and HEK293 cell lines (Fig. 1c).

To further investigate the substrates of MMR in PE, we measured the editing efficiencies of a transition (G > A), two transversions (C > G and C > T), a 1 bp insertion and a 3 bp deletion, all within a different endogenous locus, the *RNF2* locus. We found that active MMR significantly diminished the efficiency of all these edits, ranging from 1.6-fold to 14-fold, using HEK293 cells that lack MLH1, a factor that is part of both the MutLα and MutLβ heterodimers, which together repair base/base mismatches, indels of 1-2 nucleotides and larger indels (Fig. 1D). These findings were also corroborated in the MLH1/MSH3-deficient HCT116 cell line (Supplementary Fig. 1F). To test the inhibitory role of MMR on different PE strategies, we measured the efficiency of PE2, PE3 and PE3b on the *FANCF* locus, via the installation of either an A > C or a G > T substitution in HEK293 and HCT116 cells. Editing efficiency was improved by MMR deficiency for all types of PE (1.8 to 16-fold), albeit to a lesser extent for PE3 (Fig. 1e and Supplementary Fig. 1G). Overall, these results show that MMR counteracts PE efficiency across different edits and different genomic loci, in various human cell lines.

Since both HCT116 and HEC59 are cancer-derived cell lines that display MMR deficiency, it is possible that the higher levels of PE efficiency are due to cellular adaptation. The human retinal pigment epithelial-1 cell line (RPE1) is a non-cancer derived cell line, thus we utilised this for corroborating our findings. PE efficiencies are generally very low in RPE1 wild-type cells (Supplementary Fig. 1H). To overcome this shortcoming, we developed a lentivirus system for stable delivery of the PE3 system, where RPE1 cells constitutively express Cas9(H840A)-RT (denoted RPE1 PE2-BSD). We generated a CRISPR genetic knockout for the MLH1 factor in this cell line (Supplementary Fig. 1I) and performed PE3, by transducing both the pegRNA and the nicking sgRNA installing a 5 bp deletion within the *HEK3* locus. We observed an editing efficiency of approximately 35% in WT RPE1 that was further increased to 60% in RPE1-MLH1$^{-/-}$ (Fig. 1f). We additionally extended our findings to human induced-pluripotent stem cells (hiPSCs), engineered to be deficient for either MLH1 or MSH2[32] (Supplementary Fig. 1J). Wild-type hiPSCs demonstrated 20% editing efficiency of a 5 bp deletion in the *HEK3* locus, while the MLH1 and MSH2 deficient counterparts displayed an increased

efficiency of approximately 55 and 40%, respectively (Fig. 1g). Overall, these results confirm that the MMR pathway specifically plays a role in counteracting PE.

**Mismatch repair factors are recruited to sites of prime editing**. To confirm that the MMR proteins are directly involved in the processing of PE intermediates, we determined if they are recruited to sites of ongoing editing marked by Cas9(H840A)-RT. Cas9(H840A)-RT was directed to human repetitive telomeric regions, a strategy that has proven efficient for imaging Cas9[33] (Fig. 2a). Using this experimental approach, we were able to colocalize TRF1 (an essential component of the telomeric shelterin complex) with catalytically inactive Cas9 (dCas9), as previously described[34] and also with Cas9(H840A)-RT (Supplementary Fig. 2A-B). Therefore, this setup allows for the visualisation of genomic loci undergoing PE, in a pegRNA-dependent manner. Next, we used this system in U2OS expressing Green Fluorescent Protein (GFP), or GFP-tagged MMR proteins, as well as two additional DNA repair proteins that do not function in MMR (DDB2 that functions in nucleotide excision repair and 53BP1 that promotes non-homologous end-joining). We observed that 65% of MLH1-GFP foci and 25% of MSH2-GFP foci colocalised with Cas9(H840A)-RT foci (Fig. 2b, c). Importantly, we did not observe colocalisation of either DDB2-GFP or 53BP1-GFP foci and Cas9(H840A)-RT foci (Fig. 2b, c). Furthermore, by using an antibody against MLH1 (Supplementary Fig. 2C-D) we confirmed the localisation of endogenous MLH1 to sites of PE (Fig. 2d, e). We found that 30% of MLH1 foci colocalised with Cas9, while we did not observe colocalisation when a dCas9, or a sgRNA, were used (Fig. 2d, e). These findings reveal that intermediates of PE are substrates of MMR, and we propose that MMR functions to degrade the invading heterologous strand and thus restore the original DNA sequence.

**Reversible mismatch repair depletion can be exploited to increase prime editing efficiency**. We next sought to transiently deplete MLH1 as a strategy to improve PE efficiency. Since loss of MMR leads to increased mutational burden and genome instability[35], long-term inhibition of MMR is not desirable. Thus, to achieve transient MMR ablation, we depleted MLH1 in HEK293 cells with a pool of siRNAs (Supplementary Fig. 3A), and subsequently showed that this effectively increased PE efficiency by approximately 2-fold through the generation of a 5 bp deletion in the *HEK3* locus (Fig. 3a).

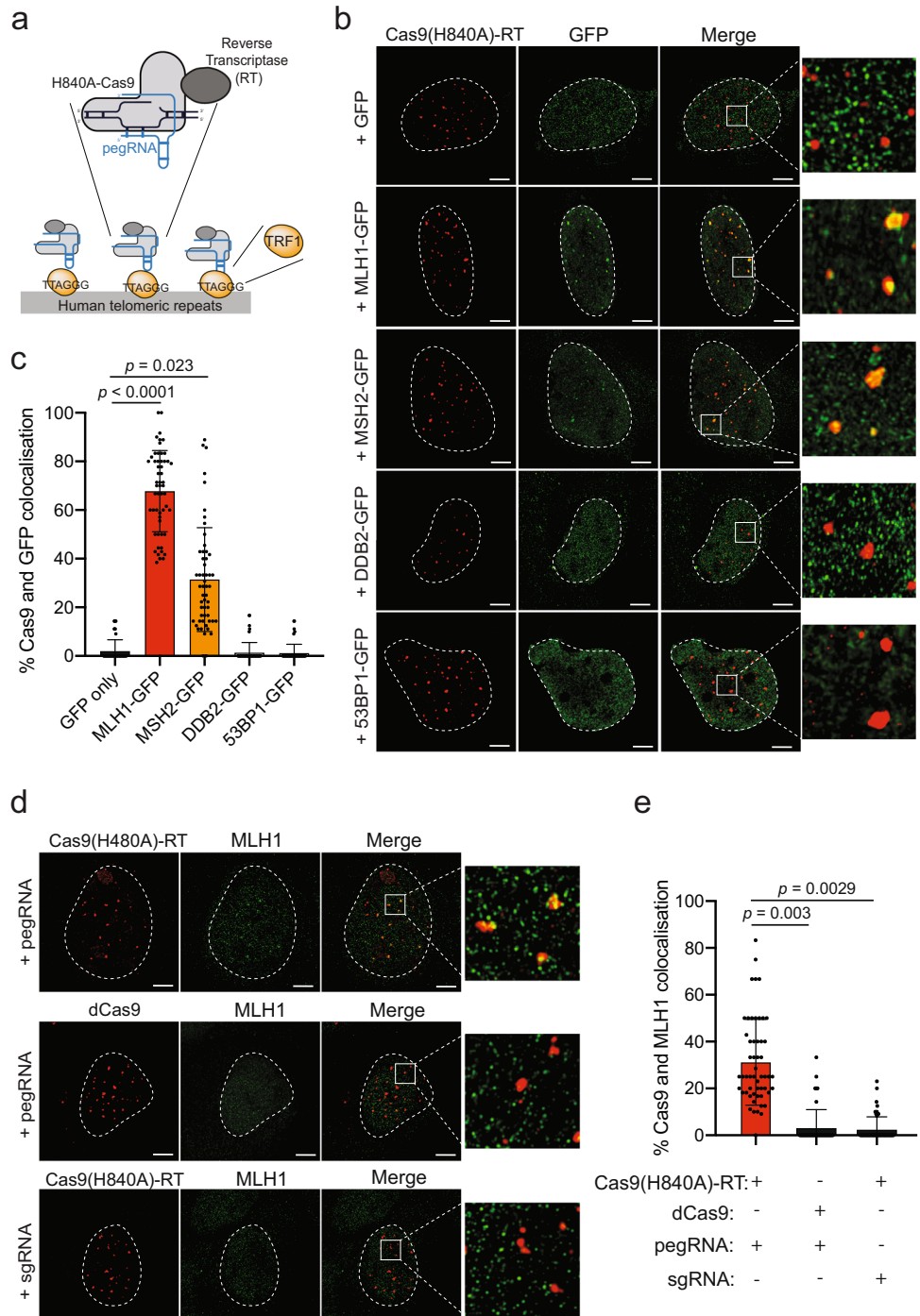

**Fig. 2 The mismatch repair protein MLH1 localises to sites of active prime editing. a** Scheme of the setup used for imaging. Cas9(H840A)-RT is targeted, through a pegRNA, to human telomeric repetitive regions. TRF1 is a telomeric protein that binds to these regions. **b** Representative super-resolution images of Cas9(H840A)-RT and the indicated GFP-tagged DNA repair proteins, in U2OS cells 24 h after reverse transfection with GFP or GFP-tagged MLH1, MSH2, DDB2, or 53BP1, as well as a pegRNA targeting telomeric repeats. $n =$ minimum 50 cells examined over three biologically independent experiments. Scale bar $= 5$ μm. **c** Quantification of **b** indicating colocalization of Cas9(H840A)-RT foci with GFP foci. $n =$ minimum 50 cells examined over three biologically independent experiments. **d** Representative super-resolution images of Cas9(H840A)-RT, or dCas9, and MLH1 with a pegRNA or a sgRNA targeting telomeric repeats. $n =$ minimum 50 cells examined over three biologically independent experiments. Scale bar $= 5$ μm. **e** Quantification of **d**, indicating colocalization of Cas9(H840A)-RT foci with MLH1 foci. $n =$ minimum 50 cells examined over three biologically independent experiments. Statistical analysis using unpaired two-tailed Student's $t$-test across biological replicates only. Error bars reflect mean ± s.e.m. Source data are provided as a Source Data file.

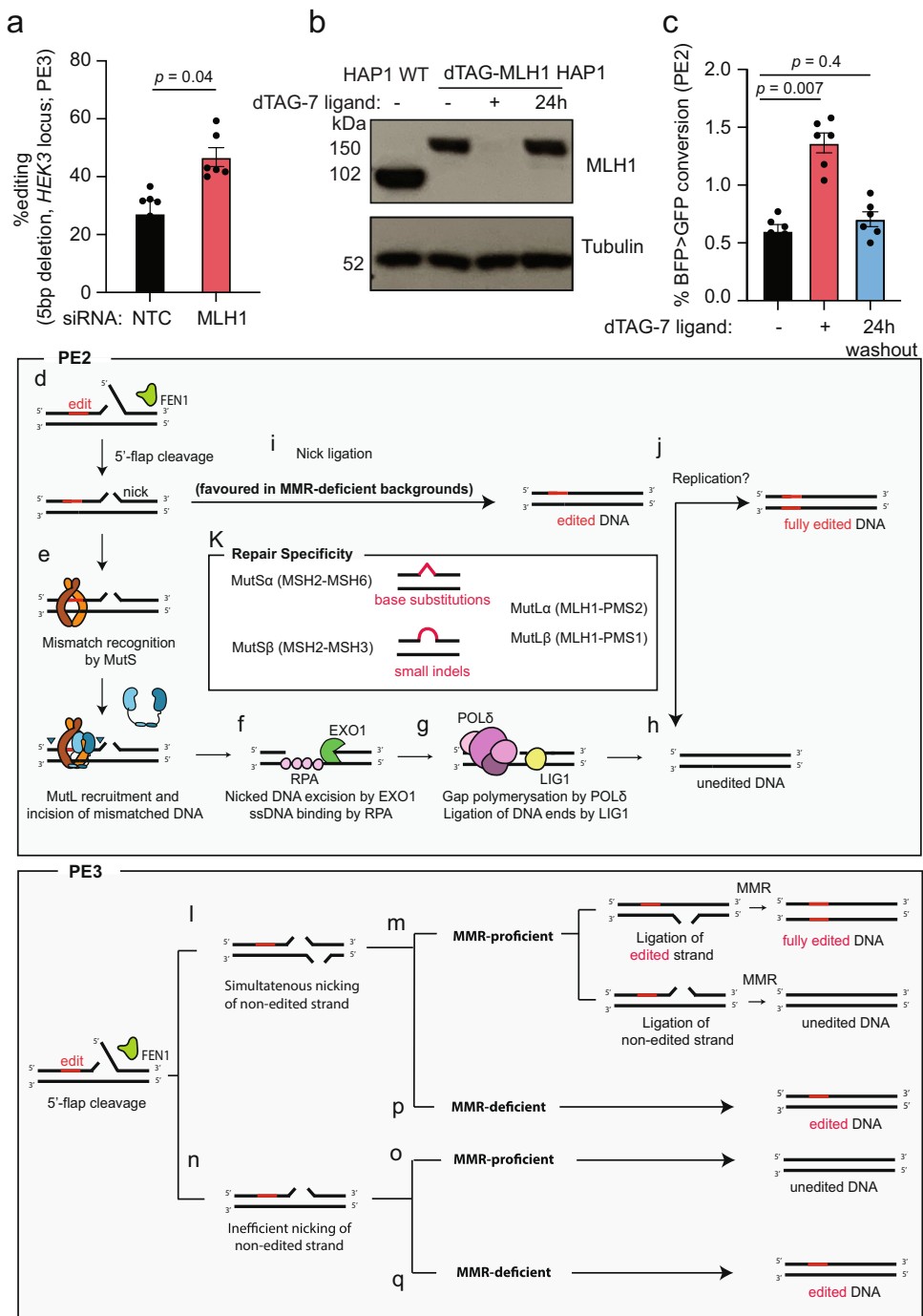

**Fig. 3 Reversible ablation of MLH1 can be exploited to increase prime editing efficiency. a** PE efficiency of a 5 bp deletion (*HEK3* locus) in HEK293 cells transfected with non-targeting control (NTC) or MLH1 siRNA pools. Editing efficiency measured by Sanger sequencing and TIDE analysis, for $n = 3$ biologically independent experiments (technical replicates also depicted). **b** Immunoblot for MLH1 and Tubulin in HAP1 cell extracts with ('+') or without ('−') dTAG-7 ligand. Recovery of dTAG-MLH1 expression was measured 24 h after ligand removal ('24 h'). $n = 2$ biologically independent experiments. **c** PE2 efficiency of BFP > GFP conversion in dTAG-MLH1 HAP1 cells, untreated ('−'), or treated with dTAG-7 ligand for 4 days ('+'), or treated with dTAG-7 ligand for 24 h, followed by its removal for 3 days ('24 h washout'). Data measured by flow cytometry for $n = 3$ biologically independent experiments (technical replicates also depicted). **d**-**q** Schematic model of MMR activity in counteracting PE efficiency. After the cleavage of the non-edited 5′-flap by the flap endonuclease (FEN1), a nick is installed in the edited strand (**d**). This nick is recognised by the MutS complex, after which MutL is recruited and catalysers incisions that flank the mismatch (**e**). Exonuclease 1 (EXO1) degrades the incised DNA and Replication protein A (RPA) coats the single-stranded DNA (ssDNA) (**f**). Polymerase δ fills the gap and Ligase 1 (LIG1) ligates the nick (**g**). This repair culminates in an unedited DNA molecule (**h**). If the nick is ligated before mismatch recognition, an heteroduplex DNA is generated, containing the edit in one of the strands (**i**). The resolution of this heteroduplex potentially relies on replication (**j**). In PE3, the non-edited strand is simultaneously nicked (**l**), which directs MMR to repair the mismatch depending on which strand is ligated first (**m**). However, it is possible that the nicking of the non-edited strand is inefficient, leading to the MMR-mediated removal of the edit (**o**). In MMR-deficient backgrounds, there is ligation of the heteroduplex without removal of the edit (**p**, **q**). Statistical analysis using unpaired two-tailed Student's *t*-test across biological replicates only. Error bars reflect mean ± s.e.m. Source data are provided as a Source Data file.

An alternative approach for achieving transient loss-of-function is through targeted protein degradation. The degradation tag (dTAG) system has proved to be an efficient strategy for rapid and transient ligand-induced targeted protein degradation[36]. Using CRISPR-mediated knock-in, we introduced the dTAG into the *MLH1* locus of HAP1 cells, which allowed for the targeted degradation of MLH1 after treatment with the dTAG ligand. Importantly, the protein levels of MLH1 were restored to those found in wild-type cells after 24 h of removal of the ligand (Fig. 3b). Using a flow cytometry-based readout, in which the pegRNA encodes a 1 bp substitution that converts the Blue-Fluorescent Protein (BFP) to GFP, we observed a 3-fold increase in BFP to GFP conversion upon treatment of the cells with the dTAG ligand and subsequent endogenous degradation of MLH1 (Fig. 3c). PE efficiency at the *HEK3* locus through a 5 bp deletion, as measured by sequencing genomic DNA, was also significantly increased by 3-fold upon treatment with the dTAG-ligand (Supplementary Fig. 3B). Importantly, this effect could be rescued to wild-type levels by removing the dTAG-ligand from the medium, thus restoring MLH1 levels (Fig. 3b, c). Taken together, these results indicate that transient ablation of MMR represents a promising strategy that can be used to increase PE efficiency.

## Discussion

Here we show that the MMR pathway counteracts PE efficiency and fidelity, across different human immortalised and induced pluripotent stem cell lines, genomic loci and edit types. Although the role of MMR in PE had not been addressed experimentally, it was hypothesised to be required for the resolution of the heteroduplex DNA, thus promoting repair of the non-edited strand by utilising the edited strand as template[11,12]. Our results provide clear evidence to the contrary, namely that the MMR system functions on the PE intermediate by degrading the invading, RT-synthesised strand to restore the original sequence. This outcome conforms to our understanding of the molecular mechanism of MMR as gleaned from in vitro systems that made use of circular heteroduplex substrates and extracts of human cells[37,38]. On these substrates, activation of MMR was strictly dependent on the presence of two factors: a mismatch and a pre-existing nick in one strand that was less than 1 kb distant. The repair process was then directed to the nicked strand. Following the extrapolation of these insights into a cellular setting, MMR, activated by misincorporated nucleotides during replication, would be initiated either by the mismatch/indel and either end of an Okazaki fragment in the lagging strand, or the 3′-end of the primer of the leading strand. During recombination of homologous but non-identical fragments, MMR would be initiated by the heterology (mismatch or indel) between the invading donor and the recipient DNA strands, with the 3′-terminus of the invading strand acting as the signal required to activate the MutL endonucleases.

We speculate that PE2 resembles the latter mechanism, whereby the RT-synthesised 3′ flap would displace the 5′ terminus of the Cas9(H840A)-RT-generated nick. This would give rise to a 5′ flap, which could be cleaved off by one of several structure-specific endonucleases (SLX1, FAN1, DNA2) and finally trimmed by flap endonuclease 1 (FEN1) (Fig. 3d). Binding of MutS at the mismatch would lead to the recruitment of MutL that generates additional incisions flanking the edit (Fig. 3e). Exonuclease 1 (EXO1) would then degrade the discontinuous strand to generate a long single-stranded gap bound by Replication Protein A (RPA) (Fig. 3f). Finally, polymerase δ (POL δ) would fill the gap and Ligase 1 (LIG1) ligate the nick (Fig. 3g). This process would result in the removal of the edit (Fig. 3h). Ligation of the nick (Fig. 3d) prior to MMR activation (Fig. 3i) would generate an heteroduplex with one edited and one non-edited strand (Fig. 3i), which would

be refractory to MMR, and would persist until replication, which would give rise to 50% progeny carrying the edit and 50% non-edited (Fig. 3j). The path in Fig. 3d, i, j would be favoured in the absence of MMR, thus accounting for the increased yield of edited alleles in MMR-deficient backgrounds.

Importantly, our data confirm the results of biochemical characterisations of the substrate specificities of MutSα and MutSβ[16–20], which showed that the former recognises preferentially base/base mismatches and indels of 1–2 nucleotides, whereas the latter binds to larger indels (Fig. 3k). Based on these findings, deletion of *MSH6* should have failed to affect the outcome of PE using a 5 bp deletion, which was indeed the case, as we report here (Fig. 1a).

Besides types of edits, different PE strategies are likely to be impacted by MMR activity to different degrees. PE3 was developed as a more efficient PE strategy, in which both edited and non-edited DNA strands are nicked (Fig. 3l). When nicks are present in both strands, the nick nearer the mismatch will be preferentially deployed by the MMR system, but the excision will destabilise the duplex and may lead to a DNA DSB, which explains the increased presence of indels in the final outcome of PE3[10]. If one strand of the heteroduplex is ligated first, MMR is directed to the nicked strand. Therefore, ligation of the edited strand directs MMR to repair the non-edited strand leading to editing on both strands of the DNA heteroduplex, whereas ligation of the non-edited strand results in an unedited DNA molecule (Fig. 3m). These outcomes rely on the assumption that the nicking sgRNA acts as efficiently as the pegRNA. This might not always be the case as some DNA molecules might have been edited and nicked by the pegRNA only (Fig. 3n). This would lead to the same outcome as PE2, which is removal of the edit by MMR (Fig. 3o). In PE3, as well as PE2, cells that lack functional MMR ligate the heteroduplex DNA without removal of the edit (Fig. 3p, q). Thus, we propose that MMR activity counteracts PE3 efficiency, as well as PE2, albeit to a lesser extent. This difference is due to the loss of a clear discrimination signal of which strand to repair, created by the nick. In PE3b, the nicking of the non-edited strand is designed to happen only after the integration of the edit. Hence, we propose that the MMR dependency of PE3b is the same as PE2.

It remains to be seen what the size limitation of PE-generated indels is, that are addressed by MMR. While our work was under revision, two other studies described the suppression of PE efficiency by MMR activity and extensively characterised the types of edits that are efficiently repaired by this pathway. Chen and colleagues showed that MMR involvement decreases with increasing indel size and that G/C to C/G edits, which form C:C mismatches, are less frequently removed by MMR factors[39]. Koeppel and colleagues systematically measured the insertion efficiency of indels ranging from 1 to 69 bp in length[40]. The authors observed an overall increase of insertion efficiency upon MMR depletion, with the greatest difference seen for indels 1–4 bp long. These results agree with the known substrate specificities of MutSα and MutSβ[16–20]. Given that one of the most promising applications of PE includes insertion, deletion or replacement of large sequences of DNA, for example to tag endogenous loci within the genome[10,41], how these lesions are processed and how their insertion efficiency can be augmented, remains of substantial future interest.

Our findings suggest that the improvement in PE efficiencies in the absence of MMR does not come at the cost of generation of undesirable indels around the edit site (Fig. 1c). However, MMR deficiency brings about a mutator phenotype, which will severely limit the utility of PE protocols that make use of long-term MMR inactivation. This deleterious outcome might be substantially reduced by interfering with MMR transiently. Our results suggest

that targeting MMR factors with siRNA or protein degradation technologies, such as proteolysis targeting chimeras (PROTAC), represent promising approaches to improve PE efficiencies. Another exciting approach would be to interfere with MMR solely at the edit site, similarly to what has been described for improving the efficiency of HDR[42,43]. While our article was under revision, two new PE strategies were described, PE4 and PE5, that rely on co-expressing dominant negative MLH1 fragments with the PE2 and PE3 machineries, respectively[39]. The authors also reported that pegRNAs encoding contiguous silent or benign mutations around the intended edit function to evade recognition and repair by MMR[39]. This strategy has the potential to improve PE efficiency without the increase in mutational burden that is associated with long-term MMR loss.

Together with recent reports[39,40], our data shed new light on the molecular mechanism of a new and highly promising genome editing technology. We have shown that the MMR pathway inhibits PE efficiency by physically localising to edit sites and promoting their reversion to non-edited sequences. However, the variability in PE observed across cell lines cannot be explained solely by the involvement of MMR and other factors, such as cell cycle stage[44] or cellular metabolism, might also be contributing factors. Hence, further studies are warranted to identify alternative cellular determinants that might limit or promote the use of this technology. The advancement in knowledge reported here can be applied to further the development of prime editors, as well as the design of novel therapeutic strategies.

## Methods

**Plasmids and oligos**. DNA oligos were obtained from Integrated DNA Technologies (IDT) unless otherwise noted. pCMV-PE2 was a gift from David Liu (Addgene plasmid # 132775). pLenti PE2-BSD was a gift from Hyongbum Kim (Addgene plasmid # 161514)[45]. pU6-pegRNA-GG-acceptor was a gift from David Liu (Addgene plasmid # 132777). PegRNAs were cloned into the pU6-pegRNA-GG-acceptor using BsaI Golden Gate assembly (NEB), following the manufacturer's instructions. sgRNAs utilised in PE3 and PE3b experiments were cloned in the lenti-sgRNA puro vector, using BsmBI Golden Gate assembly (NEB), following the manufacturer's instructions. lenti-sgRNA puro was a gift from Brett Stringer (Addgene plasmid # 104990)[46]. lenti-sgRNA neo was a gift from Brett Stringer (Addgene plasmid # 104992) and it was used to clone the sgRNA utilised in PE3 experiments in RPE1 PE2-BSD cells.

For immunofluorescence experiments, the dCas9 plasmid was a gift from David Segal (Addgene plasmid # 100091)[47] and the pSLQ1651-sgTelomere(F + E) was a gift from Bo Huang & Stanley Qi (Addgene plasmid # 51024)[33]. Additionally, the following plasmids were used: pCMVTet-eGFP-MLH1, pCMVTet-eGFP-MSH2, and pEGFP-C1-53BP1. pLenti6.3 WT GFP-DDB2 was also used and it was a gift from Dr. A. Pines[48]. pmaxGFP™ (Lonza) was used for immunofluorescence experiments, as well as to test transfection efficiency. pCRIS-PITChv2-BSD-dTAG (BRD4), used for the generation of dTAG expressing cells, was a gift from Dr. Georg Winter.

BFP-positive cells were generated using the BFP dest clone plasmid. BFP dest clone was a gift from Jacob Corn (Addgene plasmid # 71825)[49].

Sequences of sgRNA, pegRNA constructs, as well as primers for genomic DNA amplification are listed in Supplementary Data 2. The pegRNA targeting telomeres included a stem loop extension as described in ref. [33]. All plasmids for mammalian cell experiments were purified using the Plasmid Plus Midi Kit (Qiagen) or the Spin Miniprep Kit (Qiagen), both including endotoxin removal steps.

For virus production, the psPAX2 and VSV.G packaging virus were used. psPAX2 was a gift from Didier Trono (Addgene plasmid # 12260). VSV.G was a gift from Tannishtha Reya (Addgene plasmid # 14888).

PegRNAs were designed using the PrimeDesign software[50] and sgRNAs were designed using the VBC score tool[51].

**Construction of plentipegRNAPuro vector**. The plentipegRNAPuro vector was generated as follows. The lenti-sgRNA puro vector was digested with EcoRI for 2 h at 37 °C followed by digestion with BsmBI for 2 h at 55 °C and treatment with 4 µl of rSAP (NEB) for 1 h at 37 °C. The mRFP and terminator sequence present in the pU6-pegRNA-GG-acceptor were PCR amplified with a forward primer converting the BsaI cut site to BsmBI and with the reverse primer containing an EcoRI cut site. The PCR product was digested with BsmBI and EcoRI as above. The vector and digest were both purified using gel extraction using the Wizard® SV Gel and PCR Clean-Up System (Promega) and ligated using T4 ligase (NEB) for 1 h at room temperature. In order to allow for Golden gate cloning using BsmBI, the BsaI cut site present in the newly assembled vector was converted to a BsmBI cut site using the Q5 Site-Directed Mutagenesis kit (NEB).

**Lentiviral production and transduction**. Lentiviral production was achieved by plating $5 \times 10^6$ xLenti™ cells (Oxgene) in a 10 cm dish transfected one day post seeding with packaging plasmids (1 µg VSV.G, 2 µg psPAX2 and 4 µg of transfer plasmid using PEI (Sigma-Aldrich). Virus containing supernatant was collected 72 h post transfection, cleared by centrifugation and stored at −80 °C.

Cell transduction was performed using spin-infection as follows. $0.5 \times 10^6$ cells were mixed in a well of a 12-well plate with varying concentrations of supernatant containing viral particles and 8 µg/mL of polybrene (Sigma) which was then centrifuged at 1800×*g* for 30 min at 30 °C.

**Mammalian cell culture**. All cells were grown at 3% oxygen at 37 °C and routinely checked for possible mycoplasma contamination. Human HAP1 cells were obtained from Horizon Discovery and were grown in Iscove's Modified Dulbecco's Medium (IMDM) (Gibco), containing L-glutamine and 25 nM HEPES and supplemented with 10% foetal bovine serum (FBS) (Gibco) and 1% Penincillin/Streptomycin (P/S) (Sigma-Aldrich). U2OS and HEK293 cells were purchased from ATCC cell repository and cultured in DMEM (Gibco), supplemented with 10% FBS and 1% P/S. HEC59, wild-type and complemented with chromosome 2, were cultured in F12 DMEM with 10% FBS and 1% P/S. HEC59 complemented cells were cultured with 400 µg/mL of geneticin (G418, Gibco). HCT116 cells, wild-type and complemented with both chromosomes 3 and 5, were cultured with McCoy's 5A medium (Gibco), with 10% FBS and 1% P/S. HCT116 cells complemented with chromosomes 3 and 5 were cultured with 400 µg/mL geneticin (G418, Gibco) and 6 µg/ml blasticidin (Invivogen). 293T-Lα were cultured in DMEM medium (Gibco) supplemented with 10% FBS or Tet-system approved FBS (Takara Bio), 1% P/S, 100 µg/mL zeocin (Gibco) and 300 µg/mL hygromycin (Gibco). 293T-Lα were grown in doxycycline (1 µg/mL) for 7 days before any experiment, to completely deplete MLH1 expression. Doxycycline was replenished in the medium every 2 days. RPE1 cells were purchased from ATCC cell repository and cultured in F12 DMEM with 10% FBS and 1% P/S. iPSCs (WT, MLH1 and MSH2-deficient) were a gift from the Nik-Zainal lab (University of Cambridge, UK) and cultured on non-tissue culture treated plates (Stem Cell Technologies) pre-coated with 10 µg/mL Vitronectin XF (Stem Cell Technologies) in TeSR-E8 medium (Stem Cell Technologies). The medium was changed daily and the cells were passaged every 4–8 days depending on confluency using Gentle Cell Dissociation Reagent (Stem Cell Technologies). Ten micromolar of ROCKi (Stem Cell Technologies) was added to the medium whenever passaging or thawing iPSCs.

**Generation of PE2-BSD RPE1 and HEK293T cells**. RPE1 and HEK293T cells were transduced at a low multiplicity of infection with the pLenti-PE2-BSD vector and selected two days post transduction with blasticidin (10 µg/mL). Transduced cells were then single cell sorted into 96-well plates and single colonies isolated following 2–3 weeks of clonal expansion. Cas9(H840A)-RT expression was confirmed by immunoblotting.

**Generation of MLH1 isogenic knockout cell lines**. MLH1 knockouts were generated in RPE1 PE2-BSD and HEK293 cell lines by nucleofection of *S. pyogenes* Cas9 together with an in vitro transcribed sgRNA. Recombinant Cas9 containing a nuclear localisation sequence and a C-terminal 6-His tag was purchased from Integrated DNA Technologies (#1081059). The sgRNA targeting MLH1 (Supplementary Data 2) was designed utilising the VBC score tool (https://www.vbc-score.org/). T7 in vitro transcription was performed using HiScribe (NEB E2050S), using PCR-generated DNA as template, as previously described here: https://doi.org/10.17504/protocols.io.bqjbmuin.

The 4D-Nucleofector System X-Unit (Lonza) was used for nucleofection. A mixture of 30 pmol of Cas9 and 60 pmol of in vitro transcribed sgRNA was prepared in a final volume of 5 µL of Cas9 buffer (20 mM HEPES-KOH pH 7.5, 150 mM KCl, 10% glycerol) and incubated for 20 min, room-temperature. 200,000 HEK293 or RPE1 cells were centrifuged (800×*g*, 8 min), washed with PBS and resuspended in 15 µL of SF Cell-Line Solution (V4XC-2032, Lonza) or P3 Primary Cell Solution (V4XP-3032, Lonza), respectively. The Cas9-sgRNA mixture was added to the cells to a final volume of 20 µL and transferred to 16-well Nucleocuvette™ strips (Lonza). Pulse was applied utilising the CM-130 programme for HEK293 cells and EA-104 for RPE1 cells. After nucleofection, cells were left to recover for 10 min at room temperature, after which they were resuspended in 80 µL of pre-warmed medium, transferred to appropriate dishes and kept in culture.

Confirm of knock-out cell lines was performed by Sanger sequencing, through amplification of genomic DNA with appropriate primers (Supplementary Data 2). Tracking of indels by decomposition was performed by the tool TIDE[52]. For RPE1 cells, more than 90% of alleles contained an out-of-frame (+1 bp) mutation, which allowed for the use of the pooled population. HEK293 cells showed a lower frequency of out-of-frame indels, hence single cell clones were seeded by limiting dilutions into 96-well plates and a clone containing a +1 bp mutation was selected, 2–3 weeks after clonal expansion, for further studies. Abrogation of MLH1 expression was confirmed in both cell lines by immunoblotting.

**Focused DNA repair genetic screen**. CRISPR-Cas9 knockouts of DNA repair genes were generated in collaboration with Horizon Genomics. Sequences of sgRNAs were designed by Horizon Genomics or with the use of http://chopchop.cbu.uib.no/. sgRNA sequences and frameshift mutations can be found in Supplementary Data 1.

For the genetic screen, 80,000 cells were seeded in technical duplicates in 12-well plates. Cells were transfected the day after with 636 ng of pCMV-PE2 and 159 ng of the *HEK3* pegRNA inducing a 5 bp deletion, per well. 1.6 μL Lipofectamine 2000 (ThermoFisher Scientific) were used per well, following the manufacturer's instructions. A separate transfection control was performed using 795 ng of the pmaxGFP™ vector (Lonza). Medium containing transfection reagents was removed 16 h post-transfection. Transfection efficiency was measured 48 h after transfection, by determining the percentage of GFP positive cells by flow cytometry. Genomic DNA was harvested 96 h post-transfection, using the QIAmp DNA Blood Mini kit (Qiagen), following the manufacturer's instructions.

**Transfection and genomic DNA preparation of mismatch repair-deficient cell lines**. HEC59, HCT116, 293T-Lα and HEK293 cells were seeded in 48-well plates in duplicates (50,000 cells/well). Transfections were performed the next day, using 1 μL Lipofectamine 2000 (ThermoFisher Scientific) per well, following the manufacturer's instructions. Cells were transfected with 320 ng of the pCMV-PE2 vector, 80 ng of the respective pegRNA and, for PE3 and PE3b, 33.2 ng of the nicking sgRNA, per well. A transfection control was performed in parallel, by transfecting 400 ng per well of the pmaxGFP™ vector (Lonza).

iPSCs were seeded in 48-well plates in duplicates (50,000 cells/well). Transfections were performed the next day, using 1 μL Lipofectamine Stem (ThermoFisher Scientific) per well, following the manufacturer's instructions. Cells were transfected with 320 ng of the pCMV-PE2 vector, 80 ng of the respective pegRNA and, for PE3 and PE3b, 33.2 ng of the nicking sgRNA, per well. A transfection control was performed in parallel, by transfecting 400 ng per well of the pmaxGFP™ vector (Lonza).

Genomic DNA was extracted 96 h after transfection, by removing the medium, resuspending the cells in a lysis solution (100 μL DirectPCR Lysis Reagent (Cell) (Viagen Biotech), 76 μL of water and 4 μL Proteinase K) and incubating 45 min at 55 °C and 45 min at 85 °C.

**Prime editing in RPE1 cells**. Wild-type and MLH1-knockout RPE1 PE2-BSD cells were transduced at a high multiplicity of infection with the plentipegRNAPuro encoding a 5 bp deletion in the *HEK3* locus, together with a nicking sgRNA for PE3 cloned in the lenti-sgRNA neo vector (Supplementary Data 2). Spin-infection was performed with 500,000 cells/well in a 12-well plate with 8 μg/mL polybrene (1800×g, 90 min, 32 °C). Cells were selected the day after transduction with blasticidine (10 μg/mL), puromycin (2 μg/mL), and G418 (400 μg/mL). Genomic DNA was extracted as described in the section 'Transfection and genomic DNA preparation of mismatch repair-deficient cell lines', 96 h post-transduction. Antibiotic selection was maintained throughout the entire duration of the experiment. Prime editing efficiency was measured by Sanger sequencing, after amplification of the genomic DNA with appropriate primers (Supplementary Data 2). Editing efficiency was calculated by sequence decomposition, using TIDE[52].

**High-throughput DNA sequencing of genomic samples**. Genomic sites of interest were amplified from genomic DNA samples and sequenced on an Illumina Miseq or NextSeq, depending on the number of pooled samples. Amplification primers containing Illumina forward and reverse primers (Supplementary Data 2) were used for a first round of PCR (PCR1) to amplify the genomic region of interest. A mixture of staggered forward primers was used to create complexity. PCR1 reactions were performed in a final volume of 25 μL, using 0.5 μM of each forward and reverse primers, 1 μL genomic DNA and 12.5 μL of Phusion U Multiplex PCR 2× Master Mix (ThermoFisher Scientific). PCR1 was carried as following: 98 °C 2 min, 30 cycles [98 °C 10 s, 61 °C 20 s, and 72 °C 30 s], followed by a final extension of 72 °C for 7 min. Unique Illumina dual index barcode primer pairs were added to each sample in a second PCR reaction (PCR2). PCR2 was performed in a final volume of 25 μL, using 0.5 μM of each unique forward and reverse Illumina barcoding primer pair, 1 μL of unpurified PCR1 reaction and 12.5 μL of of Phusion U Multiplex PCR 2× Master Mix. PCR2 was carried as following: 98 °C 2 min, 12 cycles [98 °C 10 s, 61 °C 20 s, 72 °C 30 s], followed by a final extension of 72 °C for 7 min. PCR products were analysed by electrophoresis in a 1% (w/v) agarose gel and purified using magnetic AMPure XP beads (Beckman Coulter), using a ratio of beads:PCR product of 2:1. DNA concentration was measured by fluorometric quantification (Qubit, ThermoFisher Scientific) and sequenced on an Illumina instrument, according to manufacturer's instructions.

Sequencing reads were collected and demultiplexed using Illumina MiSeq Control software v4 (Illumina) and alignment of amplicon sequences to a reference sequence was performed using CRISPResso2[53]. CRISPResso2 was ran in standard mode and prime editing yield was calculated as: number of aligned reads containing the desired edit/total aligned reads. Percentage of indels was calculated as: number of aligned reads containing indels that are not the desired edit/ total number of aligned reads.

**siRNA transfections**. The following siRNAs from Dharmacon (used at a final concentration of 100 nM) were used in this study: MLH1 SMARTpool ON-TARGETplus (L-003906-00-0005) and Non-targeting control SMARTpool ON-TARGETplus (D-001810-10-05). siRNA transfections in HEK293 cells were performed using Dharmafect 1 following manufacturer's instructions. siRNA delivery was performed 48 h prior to transfection of prime editing vectors.

**Generation of dTAG-MLH1 HAP1 cell line**. A targeting vector encoding for the BSD-dTAG sequence (amplified from pCRIS-PITChv2-BSD-dTAG (BRD4)) surrounded by two 1 kb-long homology arms upstream and downstream of the start codon of MLH1 was generated using Gibson assembly (NEB). An in vitro transcribed sgRNA targeting the region spanning the start codon of MLH1 was generated as previously described[2]. Cas9 protein (IDT) together with the targeting vector and in vitro transcribed sgRNA were nucleofected into 200,000 haploid cells in 16-well strips, using a 4D Nucleofector (Lonza) and the programme DS-118. Three days after nucleofection, 10 μg/mL blasticidin (Invivogen) were added to the culture medium for 1 week, after which single and haploid clones were sorted into 96-well plates. Clonal haploid populations were grown and validated for correct homology-directed repair by LR-PCR and immunoblot analysis. dTAG-7 (R&D Systems), at the final concentration of 500 nM, was used to test target degradation in the generated clones and all further targeted protein degradation experiments.

**Generation of dTAG-MLH1 HAP1 BFP-positive cell line**. dTAG-MLH1 HAP1 cells were transduced at a low multiplicity of infection with the BFP dest clone plasmid. Spin-infection was performed with 500,000 cells per well in a 12-well plate (1800×g, 30 min, 30 °C). Cells were cell sorted by fluorescence (BD FACSMelody), 1 week after transduction.

**Prime editing in dTAG-MLH1 HAP1 cell line**. Twenty-five thousand dTAG-MLH1 HAP1 BFP-positive cells were seeded in two technical replicates and three biological replicates in 48-well plates, treated or not with dTAG-7 (R&D Systems) at the final concentration of 500 nM, as indicated. The day after seeding, cells were transfected with 200 ng of the pCMV-PE2 vector and 50 ng of a pegRNA cloned into the pU6-pegRNA-GG-acceptor vector, encoding a 1 bp substitution in BFP, converting it to GFP (Supplementary Data 2). dTAG-ligand was replenished in the '+' condition and removed from the '24 h' condition. Medium was replaced every 24 h for the entire course of the experiment (96 h), always replenishing dTAG-7 in the '+' condition. Prime editing efficiency was determined by percentage of GFP positive cells, measured by flow cytometry.

Prime editing efficiency of the *HEK3* locus was measured by seeding 25,000 dTAG-MLH1 HAP1 cells in two technical replicates and three biological replicates in 48-well plates, treated ('+') or not ('−') with dTAG-7 (R&D Systems) at the final concentration of 500 nM. The day after seeding, cells were transfected with 200 ng of the pCMV-PE2 vector, 50 ng of the *HEK3* pegRNA and 33.5 ng of the *HEK3* nicking sgRNA for PE3 (Supplementary Data 2), using 0.5 μL of Lipofectamine 2000 (Thermo Fisher Scientific) and following the manufacturer's instructions. Medium was replaced every 24 h for the entire course of the experiment, always replenishing dTAG-7 in the '+' condition. Genomic DNA was extracted 96 h after transfection, by removing the medium, resuspending the cells in a lysis solution (100 μL DirectPCR Lysis Reagent (Cell) (Viagen Biotech), 76 μL of water and 4 μL Proteinase K) and incubating 45 min at 55 °C and 45 min at 85 °C. Prime editing efficiency was determined by Sanger sequencing, after amplifying genomic DNA with appropriate primers (Supplementary Data 2) and measured by sequence decomposition using TIDE[52].

**Immunoblotting**. Cell extracts were prepared in RIPA lysis buffer (NEB) supplemented with protease inhibitors (Sigma) and phosphatase inhibitors (Sigma, NEB). Immunoblots were performed using standard procedures. Protein samples were separated by sodium dodecyl sulfate-polyacrylamide gel electrophoresis (SDS-PAGE) (4–12% gradient gels, Invitrogen) and subsequently transferred onto nitrocellulose membranes. Primary antibodies for MLH1 (554073, BD Pharmigen), MSH2 (ab52266, Abcam), MSH3 (ab69619, Abcam), Tubulin (3873, Cell Signalling) and ß-Actin (A5060, Sigma) were used at 1:1000. Secondary antibodies were used at 1:5000 (HRP-conjugated goat anti-mouse or anti-rabbit IgG from Jackson Immunochemicals). Immunoblots were imaged using a Curix 60 (AGFA) table-top processor. Uncropped immunoblots can be found in the Source Data file (for Fig. 3b) and Supplementary Fig. 5.

**Immunofluorescence**. U2OS cells were reverse transfected using PEI (Sigma-Aldrich). 50,000 cells were seeded per well of μ-Slide 8-well (Ibidi) chambered coverslip plates. Pre-extraction was performed using 0.1% Tween in PBS 24 h after reverse transfection. Cells were then fixed with 4% para-formaldehyde and fixed cells were processed for immunofluorescence using the following antibodies: anti-Cas9 (Cell Signalling, 14697), anti-TRF1 (Abcam, ab1423), anti-MLH1 (Thermo-Fisher, A300-015A) and anti-GFP (Abcam, ab6556). Primary antibodies were diluted 1:500 and secondary antibodies (Alexa Fluor® 568 goat anti-mouse and Alexa Fluor® 488 goat anti-rabbit, LifeTechnologies) were diluted 1:2000.

**Imaging**. Sixteen-bit fluorescence images were acquired using an Olympus IXplore spinning disk confocal microscope (equipped with the Yokogawa CSU-W1 with 50 μm pinhole disk and a Hamamatsu ORCA Fusion CMOS camera). A 60× oil immersion objective (NA 1.42) in combination with a 3.2× magnification lens (equalling 192× total magnification) was used for super-resolution imaging of fixed cells and z-stacks with a 0.24 μm slice interval were acquired. These z-stacks were then processed using the Olympus 3D deconvolution software (cellSens Dimension 3.1) (constrained iterative deconvolution, using automatic background removal and noise reduction, filter using advanced maximum likelihood algorithm and five iterations). Finally, 'maximum-z' projection images of the deconvoluted z-stacks were generated. Olympus 3D deconvolution software (cellSens Dimension 3.1) was used for analysis. Nuclear foci were counted manually and at least 50 cells per condition were imaged in each experiment. Quantification of the foci was performed manually based on maximum intensity projections.

**Statistical analysis**. Statistical parameters including exact value of $n$ (e.g., total number of experiments, measured cells), deviations, $p$ values and type of statistical test are reported in the respective figure legends. Statistical analysis was carried out using Prism 8 (GraphPad Software). Statistical analysis was performed only across biological replicates, by taking the average of the respective technical replicates, when appropriate. Error bars displayed in graphs represent the mean ± s.e.m of at least three biologically independent experiments. Statistical significance was analysed using unpaired two-tailed Student's $t$-test. $p < 0.05$ was considered significant.

**Reporting summary**. Further information on research design is available in the Nature Research Reporting Summary linked to this article.

## Data availability

The sequencing data generated in this study has been deposited in the European Nucleotide Archive (EMBL-EBI; ENA) under accession code PRJEB47501. Source data are provided with this paper.

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

## Acknowledgements
J.F.da.S. is supported by a DOC fellowship from the Austrian Academy of Sciences (ÖAW25035, awarded to J.F.da.S.). A.M. is funded by the Austrian Science Fund (Grant number P33024, awarded to J.I.L.). The Loizou lab is funded by an ERC Synergy Grant (DDREAMM Grant agreement ID: 855741, awarded to J.I.L.). This work was funded, in part, by a donation from Mr Benjamin Landesmann. CeMM is funded by the Austrian Academy of Sciences. We would like to thank the VBCF (Vienna, Austria) for all next generation sequencing. We would like to acknowledge Dr Jacob Corn (ETHZ, Zurich, Switzerland) and Dr Richard Sherwood (Harvard Medical School, Boston, USA) for critically reading the manuscript. We would like to acknowledge Dr Markus Schröder (Corn Lab, ETHZ, Zurich, Switzerland) for bioinformatic analysis. We would like to thank Mr Chris Fell (LBI-RUD/CeMM, Vienna, Austria) and Mr Marc Wiedner (CeMM, Vienna, Austria) for technical support. We are thankful to Prof KJ Patel (Weatherall Institute of Molecular Medicine, Oxford, UK) for providing the FANCD2 deficient HAP1 cell line. Prof Nik-Zainal (University of Cambridge, UK) generously provided the MLH1 and MSH2-deficient human induced pluripotent stem cells. The Jackson lab (The Gurdon Institute, University of Cambridge, UK) kindly provided the RPE1 cells. The Winter Lab (CeMM, Vienna, Austria) provided valuable tools for protein degradation. We are grateful to members of the Loizou lab for helpful discussions and feedback.

## Author contributions
J.F.daS., G.O., J.J. and J.I.L. conceptualized the study. J.F.daS. and J.I.L. obtained funding. J.F.daS., G.O., E.A., C.K. and A.M. carried out investigations. G.T. contributed to the confocal microscopy. J.F.daS. performed analysis and visualization. J.I.L. supervised the study. J.F.saS. with input from J.I.L. wrote the original draft and all authors reviewed and edited the final manuscript.

## Competing interests
The authors declare no competing interests.

## Additional information

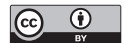

