## [Peer Review File · Nature Communications]

Reviewers' Comments:

Reviewer #1:

Remarks to the Author:

In this study, Ferreira da Silva et al show that the Mismatch Repair (MMR) pathway limits prime editing (PE) efficiency in a variety of cell lines in culture. In addition to providing insight into the cellular pathways that regulate PE, their findings offer new possibilities in the modulation of PE efficiency for genome editing purposes.

Even though the manuscript does not provide much insight into the mechanism by which MMR regulates PE, the work is timely and likely to be of high interest/value to a large number of researchers. I think the work merits publication in a high profile journal such as Nature Communication, but I suggest the authors consider the points below.

Comments:

-Abstract: the authors claim that "ablation of mismatch repair yields an up to 17-fold increase in prime editing efficiency across different human cell lines, several types of edits and multiple genomic loci." While this is true, in the vast majority of experiments a much lower improvement is observed. I think it would be better to provide the range of improvement, or remove the extreme case.

- Figure 1: Even though MMR seems to inhibit prime editing in all cell lines tested, the variation between cell lines (e.g. HCT116 vs 293T, Figure 1B) is higher than the improvement made by removing MMR. In addition, the 293T-Lalpha cell line, which over-expresses MLH1, has by far the highest PE efficiency. How do the authors interpret/explain such massive cell line-to-cell line variability? What happens in more 'normal' cell lines? Have the authors tested RPE1 cells, for example? Also, have the authors tested if over-expression of MMR components in MMR+ cell lines reduces PE further (e.g. in HEK293 cells in Figure 2E)?

- Figure 2A-C: the colocalization data should be strengthened. First, it is unclear to me why have only 13 cells been analyzed. I would expect anything between 50-100 cells as a minimal requirement in order to ensure that a reasonable representation of the cell population is achieved (asynchronous cultures contain cells at all cell cycle stages, etc). Second, I am surprised that the MLH1-Cas9(H480)-RT co-localization is anywhere between 10-70%, depending on the cell. Is there, for example, a cell cycle stage effect? Have the authors estimated how much colocalization would be expected at random? This is important given that the MLH1 signal is almost pan nuclear. siMLH1 could also be used to ensure that the MLH1 staining is specific. Have the authors tested recruitment of other MMR factors? To generalize the findings this would be desirable.

-Figure 2F-G: the MLH1-dTAG works really well, depleting the protein extremely efficiently. However, the increase in prime editing is only 2-fold, much lower than the 7-fold in mlh1Δ HAP1 cells in Figure 1A. A similar modest improvement in MMR efficiency is observed with siMLH1 (Figure 2E). Do the authors have an explanation for this? Could it be that cellular adaptation to long term loss of MMR (in all KO cells) involves the acquisition of properties that improve prime editing? Can the authors re-express MLH1 in the mlh1Δ cell line used in Figure 1 and reduce once again PE efficiency?

-Discussion: The authors 'propose that MMR proteins recognize the heteroduplex DNA, containing the desired edit introduced by the pegRNA, as a mismatch. The mismatch is promptly removed, and the non-edited strand is used as a template to restore the original sequence of nucleotides.' At this point I see this as speculative, not really substantiated by direct data. I think 'speculate' should be used instead of 'propose' before more detailed data can be used to build a molecular model of the events that take place in PE. On the other hand, I think the authors should discuss further how could PE work in the absence of MMR, which was assumed to facilitate PE.

Reviewer #2:

Remarks to the Author:

Ferreira da Silva and Olivera et al present a manuscript claiming:

1. The efficiency of prime editing is increased up to 17-fold by disruption of the mismatch repair pathway.
2. MLH1 accumulates at sites of prime editing.
3. This effect is generalizable to different cell lines and loci.

I find the ideas in the paper exciting and believe that the future of the gene editing field relies on studies like this one that precisely define the molecular events that occur during gene editing. I thus find the topic of the paper high impact and interesting to gene editing and DNA repair audiences. The scope of the work is also appropriate for publication. However, I have some concerns about the novelty of the work and the mechanism of action (these are textual points). I am also concerned about the strength of the localization data and the specificity of the effect seen in Hct116 and HEC59 cell lines. My recommendation is thus to revise the manuscript. Specific comments are below. Any experiments I suggest to address my concerns are recommendations and not requirements.

Major concerns

1. The authors do not properly define the novelty of their results in terms of prior work in this field. The base editing and prime editing manuscripts (PMID 31634902 and 29160308) clearly state that MMR is involved in both types of targeted gene editing and that a major part of the prime/base editing strategy is to nick the unedited strand to drive mismatch repair using the edited strand as the template. Here the authors conclude MMR is involved in prime editing by alternate means (a targeted genetic screen). It is not obvious to me that the Liu group has tested their MMR hypothesis, nor do they cite other studies that adequately support their model. I am therefore confused about the novelty of this manuscript. If MMR has previously been demonstrated to be involved in base/prime editing, then the current findings are not novel. If the authors have validated a heretofore untested model, then the results are novel (and very interesting). I recommend the authors clarify these points in the introduction and discussion of their manuscript. This is related to major concern 2 below. I think the authors should contextualize their work more thoroughly. Given the expertise of people in the author list, I expected the discussion of MMR and its contribution to gene editing to be much more developed.

2. Related to concern 1 above, the current manuscript does not interpret results and suggest a model consistent with these results. This is a general concern, but there were a number of specific questions that I had while reading the manuscript that would be addressed by some kind of holistic model. For example, based on the genetics, it would seem that MutS β (MSH2-MSH3) recognizes heteroduplex DNA formed between the edited nicked strand and the unedited strand. How then do the authors explain the increase in editing when EXO1 is deleted? Is the assumption that MutS-MutL cleave the edited strand 5' of the edited sequence? Additionally, do the PE3b results support or conflict with the authors' model?

3. Localization data is intriguing but unconvincing. Figure 2B shows a large amount of MLH1 signal in both + and - pegRNA samples. The MLH1 foci appear to enlarge and colocalize with Cas9(H840)-RT when pegRNA is present. My concerns are: 1) Cas9 is recognized with an antibody, thus apoCas9 (-pegRNA) may stain differently than the +pegRNA sample; 2) the effect may be specific to telomeres; and 3) only MLH1 is tested. To address these concerns, I suggest using +sgRNA (same targeting sequence but no template as the negative control), testing at least one non-telomeric locus, and testing MSH6 (which should not colocalize based on genetic results). Other experimental approaches (e.g. ChIP-qPCR) would also address my concerns with the current localization data.

4. The authors do not convince me that the MMR effect seen in Hct116 and HEC59 cells is specific to changes in the MMR context of these cells. My understanding is that these cells have MMR restored by the addition of a whole chromosome or chromosomes. How sure are the authors that the change in prime editing is specific to the missing MMR gene and they have not altered the cells in some nonspecific way (by altering transfectability, for example)? The authors do partially address this concern by using 293T-La, but the phenotype is not as strong in these cells (Figure

1B), differs in other ways (Figure S1D), and it is not used as frequently as the Hct116 and HEC59 cells. One approach to address this concern would be to show that MMR independent DNA repair events (maybe Cas9 cutting) are equivalent in +MMR and -MMR variants of a cell line. Another approach would be to specifically complement the missing repair protein(s).

Minor concerns

1. Prime editing is said to be useful for gene editing in post-mitotic cells and other cell types that are reluctant to do HDR. I would have liked to see the authors validate their results in more challenging contexts and not solely in cell lines.

December, 2021

Dear Reviewers,

Thank you for your insights while reviewing our manuscript entitled 'Prime Editing Efficiency and Fidelity are Enhanced in the Absence of Mismatch Repair' (NCOMMS-21-35482). We are pleased that our manuscript was considered to be 'exciting', 'of high interest/value to a large number of researchers' and 'high impact and interesting to gene editing and DNA repair audiences'. We found the Reviewer's suggestions to be particularly important in strengthening our manuscript and hence we undertook efforts to address all comments with the most suitable approaches and our findings can be found in the revised form of our manuscript. Below, we take you through the comments raised by the Reviewers in blue and reference to all new data in our manuscript is indicated in green.

Reviewer#1

Comments:

-Abstract: the authors claim that "ablation of mismatch repair yields an up to 17-fold increase in prime editing efficiency across different human cell lines, several types of edits and multiple genomic loci." While this is true, in the vast majority of experiments a much lower improvement is observed. I think it would be better to provide the range of improvement, or remove the extreme case.

We agree with the Reviewer and have now altered this sentence to 'depending on cell line and type of edit, ablation of mismatch repair (MMR) affords a 2-17-fold increase in prime editing (PE) efficiency' (line 29-30).

- Figure 1: Even though MMR seems to inhibit prime editing in all cell lines tested, the variation between cell lines (e.g. HCT116 vs 293T, Figure 1B) is higher than the improvement made by removing MMR. In addition, the 293T-Lalpha cell line, which over-expresses MLH1, has by far the highest PE efficiency. How do the authors interpret/explain such massive cell line-to-cell line variability?

We thank the Reviewer for making this observation. Indeed, the variability in PE efficiencies is one of the factors that motivated us to begin this study, based on the observation that cell lines such as HEK293T display high levels of editing in comparison to other transformed cell lines. Here, we report MMR status as one important factor in determining efficiency outcomes of PE, however, there are certainly other factors, and these might include cell cycle or cellular

metabolism. We refer to this in the discussion section of our revised manuscript (lines 405-409).

What happens in more 'normal' cell lines? Have the authors tested RPE1 cells, for example?

We appreciate the Reviewer's comment, as it speaks for the frequently overlooked importance of using non-cancer derived cell lines to study DNA repair, given that cancer derived cell lines frequently carry mutations in DNA damage and repair genes. Through the development of a lentiviral vector system for the delivery of the pegRNA (lines 446-456), together with constitutive expression of Cas9(H840A)-RT, we observed improved levels of prime editing in the region of 2-fold, upon loss of MLH1 in RPE1 cells (**new data - Figure 1F and Supplementary Figure 1I**). We also tested prime editing efficiency in human induced pluripotent stem cells (hiPSC) lacking either MLH1 or MSH2, where we observed a 2-3 fold increase in prime editing efficiency (**new data - Figure 1G and Supplementary Figure 1J**). These results further confirm the increased prime editing efficiency upon MMR ablation in additional cell lines that are not cancer-derived.

Also, have the authors tested if over-expression of MMR components in MMR+ cell lines reduces PE further (e.g. in HEK293 cells in Figure 2E)?

We thank the Reviewer for this query, and we agree that this could be an interesting proof-of-concept experiment to validate the role of MMR in counteracting prime editing. However, as MMR factors are highly abundant in human cells (Chang et al., 2000), increasing their expression is unlikely to translate into higher activity of the pathway. On the contrary, overexpression of MLH1, in yeast, results in a mutator phenotype (Chakraborty et al., 2018). This is especially relevant for rapidly proliferating cells (such as HEK293), as the expression of MMR factors has been shown to be increased, compared with cell lines that have longer doubling times (Marra et al., 1996). Moreover, being a multi-component system, overexpression of a single protein might not necessarily translate into higher MMR efficiency. Furthermore, in our experiments we edit an endogenous locus, which should be present in two copies (one per chromosome). We therefore expect the endogenous MMR machinery to be in excess compared with the substrate we generate during prime editing – thus suggesting that increasing MMR components may not affect prime editing.

Despite these caveats, this is an interesting experiment and hence we expressed, at different concentrations, a plasmid encoding the cDNA of MLH1 in HEK293 cells and confirmed MLH1 overexpression by immunoblotting (**Rebuttal Figure 1A**). From this titration, we chose 500 ng and 1 µg as conditions to assess prime editing efficiency, using a flow-cytometry based system, where the pegRNA encodes a 1 bp substitution that converts BFP positive cells to

GFP positive cells. We detected a small decrease in prime editing efficiency in cells transfected with the MLH1 cDNA (**Rebuttal Figure 1B**), which is in line with our proposed role of MMR activity in counteracting prime editing. However, considering the reasons stated above, as well as the very small difference observed, we have decided not to include this result in the revised manuscript but include it below for the Reviewer.

Rebuttal Figure 1: A) Immunoblot showing overexpression of MLH1, 2 days after transfection of HEK293 cells with an MLH1 cDNA-expressing plasmid. **B)** Prime editing (PE2) efficiency measured by conversion of BFP to GFP by flow-cytometry (pegRNA encodes a 1 bp substitution in BFP, converting it to GFP). Measurements were performed 3 days after prime editing in HEK293 cells transfected or not with MLH1 cDNA (n=3). *indicates p-value<0.05

- Figure 2A-C: the colocalization data should be strengthened. First, it is unclear to me why have only 13 cells been analyzed. I would expect anything between 50-100 cells as a minimal requirement in order to ensure that a reasonable representation of the cell population is achieved (asynchronous cultures contain cells at all cell cycle stages, etc).

We agree with the Reviewer and thus we have now increased the strength of our findings by ensuring that the number of quantified cells across all experiments is a minimum of 50 cells per condition (**new data - Figure 2 and Supplementary Figure 2**).

Second, I am surprised that the MLH1-Cas9(H840)-RT co-localization is anywhere between 10-70%, depending on the cell. Is there, for example, a cell cycle stage effect? Have the authors estimated how much colocalization would be expected at random? This is important given that the MLH1 signal is almost pan nuclear.

This is a very important point that we have addressed by assessing how much colocalization of MLH1 would be expected at random by using:

1. dCas9 together with a telomeric pegRNA
2. Cas9(H840A)-RT together with a sgRNA

In both conditions, prime editing should not occur and thus any colocalization between Cas9 and MLH1 would be independent of prime editing. We saw very little to no colocalization between Cas9 and MLH1 in these control experiments, which shows that the colocalization

between Cas9(H840A)RT and MLH1 is dependent on active prime editing (**new data - Figure 2D-E**).

The question of cell cycle is very pertinent. Thus, we have used markers for different phases of the cell cycle (**Rebuttal Figure 2A**). Having determined that positive (**Rebuttal Figure 2A**, yellow arrow) and negative (**Rebuttal Figure 2A**, blue arrow) cells with each of the markers can be distinguished, we next transfected cells with MLH1-GFP, Cas9(H840A)-RT and the telomeric pegRNA and stained for the cell cycle markers. We then assessed in which cell cycle phase were cells that showed colocalization between MLH1-GFP and Cas9. Using this approach, we observed that around 20% of cells that showed colocalization between Cas9(H840A) and MLH1-GFP were positive for Cyclin D1, 40% were positive for CDK2 and 60% were positive for Cyclin B1, markers of G1, S/G2 and G2/M phases, respectively (**Rebuttal Figure 2B-C**). This would suggest that MMR counteracts prime editing in G2/M predominantly compared with other cell cycle phases. While we find this observation interesting, we think it would require further validation and exploration, which will follow in a later publication. However, we provide these data below (**Rebuttal Figure 2**). Interestingly, the result is in line with a recent report suggesting that cell cycle can, to some extent, influence prime editing outcomes (Wang et al., 2021). We have included this manuscript in the discussion of our revised manuscript.

Rebuttal Figure 2: A) Representative super resolution images of cell cycle markers (Cyclin D1, CDK2, Cyclin B1). Blue arrows represent negative cells and yellow arrows positive cells. Scale bars = 20 μ m. **B)** Representative super resolution images of U2OS cells positive for colocalization between Cas9(H840A)-RT and MLH1-GFP and their respective cell cycle markers. Images were taken 24 hours following reverse transfection of Cas9(H840A)-RT in the presence of a pegRNA targeting telomeres and a MLH1-GFP expression vector from at least 50 cells per condition. Scale bars = 5 μ m. **C)** Quantification of percentage of positive cells for the different markers tested showing co-localization between Cas9(H840A)-RT and MLH1-GFP (n=3 biological replicates each with a quantification of 20 cells).

siMLH1 could also be used to ensure that the MLH1 staining is specific.

We would like to thank the Reviewer for this important point regarding antibody specificity. We have now included representative figures, as well as the quantification after staining with the MLH1 antibody, in cells treated with siRNAs for MLH1, alongside non-targeting control siRNAs (**new data - Supplementary Figure 2C-D**). These results corroborate antibody specificity. We would also like to point out that, in our revised manuscript, colocalization experiments have been extended to include a GFP-antibody, in conditions where GFP-tagged MLH1, as well as other factors, are overexpressed (**new data - Figure 2B-C**). This alternative approach corroborates our findings obtained with the MLH1 antibody against the endogenous protein.

Have the authors tested recruitment of other MMR factors? To generalize the findings this would be desirable.

This is an important remark that we have addressed by now including data showing colocalization of Cas9(H840A)-RT with the MMR factor MSH2, as well as other non-MMR DNA repair factors (DDB2 and 53BP1, as negative controls) (**new data - Figure 2B-C**). We decided to use GFP-tagged versions of the proteins mentioned above so we could assess colocalization using the same antibody. Our results show that MSH2 colocalizes with Cas9(H840A)-RT, albeit to a lesser extent compared with MLH1. Factors that are not involved in MMR do not colocalize with Cas9(H840A)-RT, generalizing the findings which go together with the genetic data we show in **Figure 1A**.

-Figure 2F-G: the MLH1-dTAG works really well, depleting the protein extremely efficiently. However, the increase in prime editing is only 2-fold, much lower than the 7-fold in *mlh1Δ* HAP1 cells in Figure 1A. A similar modest improvement in MMR efficiency is observed with siMLH1 (Figure 2E). Do the authors have an explanation for this? Could it be that cellular adaptation to long term loss of MMR (in all KO cells) involves the acquisition of properties that improve prime editing? Can the authors re-express MLH1 in the *mlh1Δ* cell line used in Figure 1 and reduce once again PE efficiency?

We thank the Reviewer for this comment. In **Figure 1A** we use PE2, but in Figure 2E (now **Figure 3A**) we use PE3 (the same is true for the dTAG approach, now **Supplementary Figure 3B**). PE3 is less affected by MMR activity as compared to PE2, since the signal for strand discrimination is lost due to the simultaneous nicking of the non-edited strand (**Figure 1E**). In the revised version of our manuscript, we now discuss this (lines 349-367) and provide a model (**new model - Figure 3L-Q**). Furthermore, the data in **Figure 1A** is from Next Generation Sequencing, whereas the data from the dTAG and siRNA experiments is from Sanger sequencing, hence we propose that this explains these differences with regards to absolute efficiencies (since Sanger-based approaches are considerably less sensitive in our

experience). We have now revised the figure legends, the legends of the graph axis, as well as the main text to make these experimental differences clearer.

Regarding the rescue experiment of re-expressing MLH1 in an MLH1-deficient background, to test for cellular adaptation, we provide the following **new data in Figure 3C**. Here, we rescued MLH1 expression by withdrawing the dTAG ligand, leading to restoration of MLH1 protein. In these conditions we tested prime editing efficiency through conversation of BFP to GFP and found this to be restored to levels comparable to wild-type cells (**new data in Figure 3B-C**). This data shows that re-expression of MLH1 in an MLH1-deficient background reduces PE levels back to levels found in wild-type cells.

-Discussion: The authors 'propose that MMR proteins recognize the heteroduplex DNA, containing the desired edit introduced by the pegRNA, as a mismatch. The mismatch is promptly removed, and the non-edited strand is used as a template to restore the original sequence of nucleotides.' At this point I see this as speculative, not really substantiated by direct data. I think 'speculate' should be used instead of 'propose' before more detailed data can be used to build a molecular model of the events that take place in PE. On the other hand, I think the authors should discuss further how could PE work in the absence of MMR, which was assumed to facilitate PE.

We thank the Reviewer for pointing this out, which prompted us to address this concern and additionally expand our Discussion, leading to what we think is a much-improved manuscript. We have changed the word 'propose' to 'speculate' (line 325) and alongside the expanded Discussion, we have included a model (**new Figure 3D-Q**). This includes our hypothesis on the role of MMR in prime editing, including different prime editing strategies (PE2 and PE3), as well as how prime editing would function in MMR proficient and deficient backgrounds (lines 325-367). During the revision of our manuscript, similar findings were reported by the Liu lab (Chen et al., 2021) that we have added to our discussion to further enrich it.

Reviewer #2 (Remarks to the Author):

1. The authors do not properly define the novelty of their results in terms of prior work in this field. The base editing and prime editing manuscripts (PMID 31634902 and 29160308) clearly state that MMR is involved in both types of targeted gene editing and that a major part of the prime/base editing strategy is to nick the unedited strand to drive mismatch repair using the edited strand as the template. Here the authors conclude MMR is involved in prime editing by alternate means (a targeted genetic screen). It is not obvious to me that the Liu group has tested their MMR hypothesis, nor do they cite other studies that adequately support their

model. I am therefore confused about the novelty of this manuscript. If MMR has previously been demonstrated to be involved in base/prime editing, then the current findings are not novel. If the authors have validated a heretofore untested model, then the results are novel (and very interesting). I recommend the authors clarify these points in the introduction and discussion of their manuscript. This is related to major concern 2 below. I think the authors should contextualize their work more thoroughly. Given the expertise of people in the author list, I expected the discussion of MMR and its contribution to gene editing to be much more developed.

We would like to thank the Reviewer for raising this important point, which was partially shared with Reviewer 1 in the above point. Strategies to improve the efficiency of prime editing include approaches that nick the non-edited strand, thus directing MMR to repair the lesion using the edited strand as template (Anzalone et al., 2019). This strategy builds on a speculative model, where MMR is proposed to resolve the heteroduplex that is created during the introduction of the edit by promoting the copy of the installed edit to the non-edited strand. Nonetheless, the role of MMR was not experimentally addressed in this model, to the best of our knowledge. There are several reports that hypothetically support this model (Petri et al., 2021; Scholefield and Harrison, 2021).

Here, we report that MMR activity counteracts prime editing and thus its ablation increases its efficiency. At the time of submission of our manuscript, it was the first report of an experimental system that had tested the effect of MMR on prime editing efficiencies. It is worth noting that during the review of our manuscript the Liu lab published a manuscript revealing that MMR impedes prime editing and thus expression of a MMR-inhibiting protein can be used as a strategy to increase its efficiency. This led to the development of PE4 and PE5 (Chen et al., 2021). We have discussed these findings in our revised manuscript.

Our results do not contradict the hypothesis that nicking of the non-edited strand, in PE3, promotes edit integration through MMR activity. But they propose that MMR acts by excising the edit before the ligation of the edited strand occurs. Our work therefore helps explain why MMR counteracts prime editing efficiency to different extents depending on the strategy used (PE2 or PE3). This is further explained in our new model (**new Figure 3D-Q**), that is described in detail in the Discussion (lines 325-367). We have also contextualized the role of MMR in genome editing technologies (lines 77-81 and 307-312). We find that these additions and clarifications have improved the text and so we thank the Reviewer for raising this comment.

2. Related to concern 1 above, the current manuscript does not interpret results and suggest a model consistent with these results. This is a general concern, but there were a number of

specific questions that I had while reading the manuscript that would be addressed by some kind of holistic model. For example, based on the genetics, it would seem that MutS β (MSH2-MSH3) recognizes heteroduplex DNA formed between the edited nicked strand and the unedited strand. How then do the authors explain the increase in editing when EXO1 is deleted? Is the assumption that MutS-MutL cleave the edited strand 5' of the edited sequence? Additionally, do the PE3b results support or conflict with the authors' model?

We agree with the Reviewer, hence, this comment prompted us to propose a model of how we hypothesize MMR to function during prime editing, which covers different prime editing strategies (PE2 and PE3), as well as outcomes depending on MMR-proficiency or deficiency. We find that this model (**new Figure 3D-Q**), together with its detailed description in the Discussion section (lines 325-367) will help the reader to better understand and interpret our results.

Regarding the Reviewer's specific question on recognition of the heteroduplex DNA formed between the edited nicked strand and the unedited strand, in our genetic screen (**Figure 1A**), a 5 bp deletion was introduced by prime editing. We expect edits of this type to be recognized by the MutS β (MSH2-MSH3), but not by the MutS α (MSH2-MSH6) complex. This is because MutS β recognises small indels (such as a 5 bp deletion), whereas MutS α recognises bases substitutions and smaller indels. Indeed, our results prove this hypothesis to be correct, as MSH6 was the MMR factor whose depletion least impacted prime editing efficiency. We hypothesize that after recognition of the edit by the MutS complex, MutL is recruited, catalyzing incisions of the heteroduplex that are guided to the strand nicked by Cas9(H840A)-RT (**Figure 3E**). This is also supported by our genetic screen, where we show that depletion of either MLH1 or PMS2, both members of the MutL α complex, strongly reduced edit removal (**Figure 1A**). EXO1 is a 5' to 3' endo/exonuclease that has been shown to be required for mismatch excision in conditions where this mismatch is flanked by a 3' pre-existing nick, as well as 5' nick created by the MutL complex (Tishkoff et al., 1997; Constantin et al., 2005). Consistent with this role, we predict EXO1 to intervene in prime editing by removing the mismatched strand via its exonucleolytic activity (**new Figure 3F**). This is in line with the increased prime editing efficiency that we observe in the HAP1- Δ EXO1 cell line (**Figure 1A**). Interestingly, in a study published while our article was under revision, EXO1 depletion also led to increased prime editing efficiency, albeit to a lesser extent when compared to MutS and MutL factors, thus corroborating our results (Chen et al., 2021). We have included this study in the discussion and interpretation of our data.

Regarding the MMR dependency of other prime editing strategies, such as PE3 and PE3b, we have addressed this comment by extending our Discussion to more thoroughly interpret our results (lines 349-367) and we have included a model (**new Figure 3L-Q**).

3. Localization data is intriguing but unconvincing. Figure 2B shows a large amount of MLH1 signal in both + and – pegRNA samples. The MLH1 foci appear to enlarge and colocalize with Cas9(H840)-RT when pegRNA is present. My concerns are: 1) Cas9 is recognized with an antibody, thus apoCas9 (-pegRNA) may stain differently than the +pegRNA sample; 2) the effect may be specific to telomeres; 3) only MLH1 is tested. To address these concerns, I suggest using +sgRNA (same targeting sequence but no template as the negative control), testing at least one non-telomeric locus, and testing MSH6 (which should not colocalize based on genetic results). Other experimental approaches (e.g. ChIP-qPCR) would also address my concerns with the current localization data.

We thank the Reviewer for this important point, that was again partially shared with Reviewer 1 above. Below we go through the three points raised:

1. We here show that Cas9 staining is comparable in cells transfected with and without a pegRNA with both conditions displaying pan-nuclear Cas9 (**Rebuttal Figure 3A**). Furthermore, detection of Cas9 by immunofluorescence using antibodies that recognize Cas9 targeted to specific genomic regions has been extensively used in the literature (Chen et al., 2013; Tsouroula et al., 2016; Jayavaradhan et al., 2019). In addition, we have now performed two experiments that address this point:

- A. We have expressed dCas9, instead of Cas9(H840A)-RT, together with the pegRNA targeting telomeres (**new data Figure 2D-E**)
- B. We have expressed Cas9(H840A)-RT together with a sgRNA, instead of a pegRNA (**new data Figure 2D-E**)

In both these experiments, prime editing should not take place and we did not observe colocalization between Cas9 and MLH1, suggesting that any colocalization observed is specific to prime editing.

2. Cas9 foci have been reported to be detected upon targeting repetitive regions (Chen et al., 2013). We chose telomeres since these are highly repetitive and abundant in cells thus yielding the clear visualization we needed for imaging-based approaches. To look at other repetitive regions in the genome, we attempted to target centromeric regions. However, we did not observe a high frequency of cells containing Cas9 foci with this setup. Moreover, colocalization with a centromeric protein (CENPB) was low (**Rebuttal Figure 3B**). This indicates that this is not ideal for measuring Cas9 localization. This can be for various reasons.

Firstly, centromeric repeats in human cells contain divergent sequences, contrary to telomeric repeats that are represented specifically by TTAGGG repeats, therefore, potentially reducing the number of targeting regions. Secondly, centromeres represent regions of the genome with highly condensed chromatin which may be inaccessible to Cas9. Finally, the pegRNA we used may work less efficiently than the telomeric repeat pegRNA, which further decreases the clear visualization of foci.

3. We also tested MSH2 colocalization with Cas9(H840A)-RT and we observe a significant colocalization that corroborates the genetic data for MSH2 (**Figure 1A, new data - Figure 2B-C**). In addition, we tested 53BP1 and DDB2, two DNA repair proteins that function in different DNA repair pathways, in which we show absence of colocalization with Cas9 (**new data- Figure 2B-C**).

Rebuttal Figure 3: A) Representative super-resolution images of Cas9(H840A)-RT in U2OS cells. Images were taken 24 hours following reverse transfection of Cas9(H840A)-RT in the absence or presence of a pegRNA targeting the *HEK3* locus. Scale bars = 5 μ m **B)** Representative super-resolution images of Cas9(H840A)-RT in U2OS cells. Images were taken 24 hours following reverse transfection of Cas9(H840A)-RT in the presence of a pegRNA targeting a centromeric repeat consensus sequence. Quantification shows number of Cas9 foci that colocalize with CENPB foci, a centromeric protein..Scale bars = 5 μ m

4. The authors do not convince me that the MMR effect seen in Hct116 and HEC59 cells is specific to changes in the MMR context of these cells. My understanding is that these cells have MMR restored by the addition of a whole chromosome or chromosomes. How sure are the authors that the change in prime editing is specific to the missing MMR gene and they

have not altered the cells in some nonspecific way (by altering transfectability, for example)? The authors do partially address this concern by using 293T-L α , but the phenotype is not as strong in these cells (Figure 1B), differs in other ways (Figure S1D), and it is not used as frequently as the Hct116 and HEC59 cells. One approach to address this concern would be to show that MMR independent DNA repair events (maybe Cas9 cutting) are equivalent in +MMR and -MMR variants of a cell line. Another approach would be to specifically complement the missing repair protein(s).

We thank the Reviewer for raising this important point. Historically, long-term correction of MMR deficiency by expression of cDNA has been largely unsuccessful, with the notable exception of 293T-L α cells, used in this study. This is because, in stable clones, expression was shown to diminish with passaging, which implies silencing by DNA methylation. A fragment of chromosome 3 and chromosome 2 were used to correct the MMR defects in HCT116 and HEC59 cells, respectively, as shown in the laboratories of Paul Modrich, Tom Kunkel, among others (Koi et al., 1994; Umar et al., 1997; Chen et al., 2001; Haugen et al., 2008). These cell lines have been used in numerous assays and in many different laboratories with no anomalous phenotypes reported to our knowledge.

Nonetheless, we appreciate the Reviewer's concern and thus we tested the transfectability of HCT116 and HEC59, as compared to their complemented counterparts. We did not observe any differences (**new data - Supplementary Figure 1E**).

Additionally, as suggested by the Reviewer, we tested the frequency of MMR-independent events after Cas9 cutting. For this, we measured indels generated after nucleofecting the MMR-deficient cell lines, and respective MMR-proficient counterparts, with a sgRNA targeting the *LBR* gene. We did not observe a significant difference in the repair events after Cas9-cutting in wild-type *versus* chromosome-complemented cell lines (**Rebuttal Figure 4**).

Moreover, in order to further strengthen our findings that the change in PE efficiency is specific to MMR deficiency, we have now expanded **Figure 1** to include MMR isogenic knockouts in HEK293 and RPE1 cell lines, as well as human iPSCs (**new data - Figure 1B-G**). Taken together, we have reproduced the findings from HCT116 and HEC59 cells in different cell lines and using different systems, summarized below:

1. 293T-L α – **Figure 1B** (isogenic inducible system for MLH1 repression)
2. HEK293 – **new data - Figure 1B, D, E** (isogenic CRISPR engineered knockout for MLH1) and siRNA for MLH1 (**Figure 3A**)
3. RPE1 – **new data - Figure 1F** (isogenic CRISPR engineered knockout for MLH1)

4. human iPSCs – **new data - Figure 1G** (isogenic CRISPR engineered knockout for MLH1 and MSH2)
5. HAP1 cells – **Figure 1A** (isogenic CRISPR engineered knockouts for MLH1, PMS2, MSH2, EXO1, MSH3, MSH6). Also, a reversible dTAG system for degradation of MLH1 where re-expression of MLH1 restores editing levels to wild-type cells (**new data - Figure 3B-C**).

Rebuttal Figure 4: MMR independent DNA repair events in HCT116 and HEC59, as well as their MMR-proficient complemented counterparts. Measured by Sanger sequencing after nucleofection of the cell lines with a sgRNA targeting the *LBR* gene (n=2).

Minor concerns

1. Prime editing is said to be useful for gene editing in post-mitotic cells and other cell types that are reluctant to do HDR. I would have liked to see the authors validate their results in more challenging contexts and not solely in cell lines.

We agree with this remark. In order to expand our findings to more challenging systems, we studied the role of MMR during prime editing in human induced pluripotent stem cells (hiPSCs). For this, we used MLH1 and MSH2-deficient hiPSCs where we observed a 2-3-fold increase in PE efficiency when MMR is abolished in both mutants used (**new data - Figure 1G and Supplementary Figure 1J**).

The inhibition of MMR to increase prime editing efficiency in hard-to-edit cells is certainly of value and something we would like to explore in future studies, in part to understand cell-type specificity. As of today, in addition to cell lines and hiPSCs, manipulation of MMR has proven useful to increase prime editing efficiency in primary human T cells. This was reported in an article published by the Liu lab while our manuscript was under revision (Chen et al., 2021).

We have discussed the approaches used in this article to perturb MMR to enhance PE in the revised version of our manuscript.

References:

- Anzalone, A. V., Randolph, P. B., Davis, J. R., Sousa, A. A., Koblan, L. W., Levy, J. M., et al. (2019). *Search-and-replace genome editing without double-strand breaks or donor DNA*. doi:10.1038/s41586-019-1711-4.
- Chakraborty, U., Dinh, T. A., and Alani, E. (2018). Genomic Instability Promoted by Overexpression of Mismatch Repair Factors in Yeast: A Model for Understanding Cancer Progression. *Genetics* 209, 439–456. doi:10.1534/GENETICS.118.300923.
- Chang, D. K., Ricciardiello, L., Goel, A., Chang, C. L., and Boland, C. R. (2000). Steady-state Regulation of the Human DNA Mismatch Repair System *. *J. Biol. Chem.* 275, 18424–18431. doi:10.1074/JBC.M001140200.
- Chen, B., Gilbert, L., Cimini, B., Schnitzbauer, J., Zhang, W., Li, G., et al. (2013). Dynamic imaging of genomic loci in living human cells by an optimized CRISPR/Cas system. *Cell* 155, 1479–1491. doi:10.1016/J.CELL.2013.12.001.
- Chen, P. J., Hussmann, J. A., Yan, J., Knipping, F., Ravisankar, P., Chen, P.-F., et al. (2021). Enhanced prime editing systems by manipulating cellular determinants of editing outcomes. *Cell* 184, 5635-5652.e29. doi:10.1016/J.CELL.2021.09.018.
- Chen, S., Bigner, S. H., and Modrich, P. (2001). High rate of CAD gene amplification in human cells deficient in MLH1 or MSH6. *Proc. Natl. Acad. Sci. U. S. A.* 98, 13802. doi:10.1073/PNAS.241508098.
- Constantin, N., Dzantiev, L., Kadyrov, F. A., and Modrich, P. (2005). Human Mismatch Repair: RECONSTITUTION OF A NICK-DIRECTED BIDIRECTIONAL REACTION. *J. Biol. Chem.* 280, 39752–39761. doi:10.1074/JBC.M509701200.
- Haugen, A. C., Goel, A., Yamada, K., Marra, G., Nguyen, T. P., Nagasaka, T., et al. (2008). Genetic instability caused by loss of MutS homologue 3 in human colorectal cancer. *Cancer Res.* 68, 8465–8472. doi:10.1158/0008-5472.CAN-08-0002.
- Jayavaradhan, R., Pillis, D. M., Goodman, M., Zhang, F., Zhang, Y., Andreassen, P. R., et al. (2019). CRISPR-Cas9 fusion to dominant-negative 53BP1 enhances HDR and inhibits NHEJ specifically at Cas9 target sites. *Nat. Commun.* 10. doi:10.1038/S41467-019-10735-7.
- Koi, M., Umar, A., Chauhan, D. P., Cherian, S. P., Carethers, J. M., Kunkel, T. A., et al. (1994). Human Chromosome 3 Corrects Mismatch Repair Deficiency and Microsatellite

- Instability and Reduces N-Methyl-N'-nitro-N-nitrosoguanidine Tolerance in Colon Tumor Cells with Homozygous hMLH1 Mutation. *Cancer Res.* 54, 4308–4312.
- Marra, G., Chang, C., Laghi, L., Chauhan, D., Young, D., and Boland, C. (1996). Expression of human MutS homolog 2 (hMSH2) protein in resting and proliferating cells. *undefined*.
- Petri, K., Zhang, W., Ma, J., Schmidts, A., Lee, H., Horng, J. E., et al. (2021). CRISPR prime editing with ribonucleoprotein complexes in zebrafish and primary human cells. *Nat. Biotechnol.* 2021, 1–5. doi:10.1038/s41587-021-00901-y.
- Scholefield, J., and Harrison, P. T. (2021). Prime editing – an update on the field. *Gene Ther.* 2021 287 28, 396–401. doi:10.1038/s41434-021-00263-9.
- Tishkoff, D. X., Boerger, A. L., Bertrand, P., Filosi, N., Gaida, G. M., Kane, M. F., et al. (1997). Identification and characterization of *Saccharomyces cerevisiae* EXO1, a gene encoding an exonuclease that interacts with MSH2. *Proc. Natl. Acad. Sci. U. S. A.* 94, 7487–7492. doi:10.1073/PNAS.94.14.7487.
- Tsouroula, K., Furst, A., Rogier, M., Heyer, V., Maglott-Roth, A., Ferrand, A., et al. (2016). Temporal and Spatial Uncoupling of DNA Double Strand Break Repair Pathways within Mammalian Heterochromatin. *Mol. Cell* 63, 293–305. doi:10.1016/J.MOLCEL.2016.06.002.
- Umar, A., Koi, M., Risinger, J. I., Glaab, W. E., Tindall, K. R., Kolodner, R. D., et al. (1997). Correction of hypermutability, N-Methyl-N'-nitro-N-nitrosoguanidine resistance, and defective dna mismatch repair by introducing chromosome 2 into human tumor cells with mutations in MSH2 and MSH6. *Cancer Res.* 57, 3949–3955.
- Wang, Q., Liu, J., Janssen, J. M., Tasca, F., Mei, H., and Gonç Alves, M. A. F. V (2021). Broadening the reach and investigating the potential of prime editors through fully viral gene-deleted adenoviral vector delivery. *Nucleic Acids Res.* 49, 11986–12001. doi:10.1093/NAR/GKAB938.

Reviewers' Comments:

Reviewer #1:

Remarks to the Author:

The authors have done an excellent job addressing the comments from the reviewers. The work is ready for publication.

Reviewer #2:

Remarks to the Author:

The authors have addressed all of my comments and I recommend publication. I find the addition of figures 1F, 1G, and 3C to be especially compelling. Furthermore, the revised discussion of mechanism is great and, in my opinion, will drive further mechanistic insights. Congratulations to the authors.

Reviewer #1 (Remarks to the Author):

The authors have done an excellent job addressing the comments from the reviewers. The work is ready for publication.

Reviewer #2 (Remarks to the Author):

The authors have addressed all of my comments and I recommend publication. I find the addition of figures 1F, 1G, and 3C to be especially compelling. Furthermore, the revised discussion of mechanism is great and, in my opinion, will drive further mechanistic insights. Congratulations to the authors.

We are delighted that both Reviewers consider our work now ready for publication in *Nature Communications*. We would like to thank both Reviewers for their constructive feedback throughout the review process, that has improved our manuscript greatly.